# Metallothionein I/II Expression and Metal Ion Levels in Correlation with Amyloid Beta Deposits in the Aged Feline Brain

**DOI:** 10.3390/brainsci13071115

**Published:** 2023-07-22

**Authors:** Emmanouela P. Apostolopoulou, Nikolaos Raikos, Ioannis Vlemmas, Efstratios Michaelidis, Georgia D. Brellou

**Affiliations:** 1Department of Pathology, Faculty of Health Sciences, School of Veterinary Medicine, Aristotle University of Thessaloniki, 54627 Thessaloniki, Greece; emmaapos@vet.auth.gr (E.P.A.); ivlemmas@vet.auth.gr (I.V.); 2Department of Forensic Medicine & Toxicology, Faculty of Health Sciences, School of Medicine, Aristotle University of Thessaloniki, 54124 Thessaloniki, Greece; raikos@auth.gr; 3Laboratories of the 3rd Army Veterinary Hospital, Chemical Department, 57001 Thessaloniki, Greece; mixeusart@yahoo.gr

**Keywords:** aging, feline, metallothionein I/II, iron, zinc, Perl’s/DAB, amyloid beta, neurodegeneration, IHC, ICP-MS

## Abstract

Brain aging has been correlated with high metallothionein I-II (MT-I/II) expression, iron and zinc dyshomeostasis, and Aβ deposition in humans and experimental animals. In the present study, iron and zinc accumulation, the expression of MT-I/II and Aβ42, and their potential association with aging in the feline brain were assessed. Tissue sections from the temporal and frontal grey (GM) and white (WM) matter, hippocampus, thalamus, striatum, cerebellum, and dentate nucleus were examined histochemically for the presence of age-related histopathological lesions and iron deposits and distribution. We found, using a modified Perl’s/DAB method, two types of iron plaques that showed age-dependent accumulation in the temporal GM and WM and the thalamus, along with the age-dependent increment in cerebellar-myelin-associated iron. We also demonstrated an age-dependent increase in MT-I/II immunoreactivity in the feline brain. In cats over 7 years old, Aβ immunoreactivity was detected in vessel walls and neuronal somata; extracellular Aβ deposits were also evident. Interestingly, Aβ-positive astrocytes were also observed in certain cases. ICP-MS analysis of brain content regarding iron and zinc concentrations showed no statistically significant association with age, but a mild increase in iron with age was noticed, while zinc levels were found to be higher in the Mature and Senior groups. Our findings reinforce the suggestion that cats could serve as a dependable natural animal model for brain aging and neurodegeneration; thus, they should be further investigated on the basis of metal ion concentration changes and their effects on aging.

## 1. Introduction

Advances in veterinary care and nutrition have led to significantly longer life expectancy for pet cats, with a reported median longevity of 14 years [1]. Aging is a biological phenomenon characterized by progressive and irreversible deterioration of physiological function, which eventually leads to age-related diseases. It has also been described as an important risk factor for many neurodegenerative diseases in humans, including Alzheimer’s disease (AD) and Parkinson’s disease (PD) [2]. The brain is probably the most vulnerable tissue affected by aging. The high oxygen requirement, iron storage capacity, elevated polyunsaturated fatty acid content, and low synthesis capacity of endogenous antioxidants, along with its limited regeneration capability, lead to increased brain vulnerability to oxidative stress [3]. Studies that have focused on aging in several animal species, such as cats, have described brain lesions comparable to those observed in human brain aging and AD [4,5,6,7,8]. However, further research on age-related central nervous system (CNS) pathology is necessary to be conducted [4].

Common features of normal aging include increased blood–brain barrier permeability and glial senescence, which leads to age-dependent loss of function of neuroglia [9,10,11,12]. These changes might lead to the dysregulation of metal concentrations in the aging brain. Metal toxicity gives rise to oxidative stress, mitochondrial dysfunction, and DNA damage and may subsequently contribute to advanced aging and promote the onset of neurodegenerative diseases, such as AD and PD [13].

Iron is the most abundant essential trace metal in the brain and is vital for pleiotropic biological processes, including neurotransmitter synthesis, myelination of neurons, and mitochondrial function [14]. Similarly, zinc—the second most abundant metal in the brain—exerts modulatory effects on multiple cellular processes, such as neurotransmission, gene regulation, and enzymatic activity [15]. In normal aging, increased iron concentration can be observed in discrete brain regions: the substantia nigra, lentiform and caudate nuclei, globus pallidus, and cortices, in humans and mice [11,14,16]. Moreover, an age-dependent increase in zinc concentration has been reported in rats, humans, and mice [11]. Higher levels of iron and excessive zinc levels can trigger the generation of reactive oxygen species (ROS) [17]. Additionally, aging is associated with the inefficiency of antioxidant defense systems. Thus, elevated ROS concentrations evoke oxidative stress and lead to cell damage and eventually cell death [13,14,18]. Consequently, iron and zinc dyshomeostasis might play a critical pathophysiological role in age-related neurodegenerative diseases [13,14,19].

Taking into consideration the high zinc brain content in humans, the regulation of metal homeostasis is crucial [15]. Among other proteins, metallothioneins (MTs) are responsible for zinc regulation. They are small, cysteine-rich metal-binding proteins, which are present in most cells of the body, including several cell types in the brain and spinal cord [20]. Three MT isoforms are expressed in the mammalian brain: MT-I, MT-II, and MT-III. Of these, MT-I and MT-II (hereafter MT-I/II) have been considered equivalent proteins. They share similar expression profiles and structures, differing only by one amino acid and their ability to bind divalent metals, whereas they exert the same function [21]. Within the CNS, MT-I/II is mainly expressed in astrocytes, while other cell types also show MT-I/II expression, such as neurons, endothelial cells, leptomeningeal cells (LCs), ependymal cells, and CP epithelial cells [20,21,22]. Regarding brain aging, previous research has demonstrated elevated levels of MT-I/II in the astroglia of dogs, rats, and cattle [23,24,25,26]. Specifically, Mocchegiani and colleagues, in 2001, proposed MT-I/II as a potential marker of aging. Additionally, increased astroglial numbers are observed with age, and there is also a higher expression of astrocytic immunohistochemical markers, such as GFAP [2,4,11,14,27,28]. Earlier studies have demonstrated that high concentrations of zinc, copper, and iron are present in insoluble amyloid plaques [14,15]. Amyloid β protein has been shown to bind metal ions with high affinity, while iron and zinc exposure induces the aggregation and deposition of Aβ proteins in humans [15,29]. Aβ plaques have also been associated with MT-I/II in the hippocampus of experimental animal models of AD [21]. Additionally, the activation of astrocytes occurs as a result of Aβ load, which triggers a secondary astroglial response. An increased number of astrocytes surround Aβ plaques and, given the capability of astroglia to secrete even minor quantities of amyloid, astrogliosis may contribute alongside neurons to further Aβ production [30].

Human and animal brain aging has been associated with iron and zinc dyshomeostasis [11,13,14,16,17,26], as well as with MT-I/II expression [31]. The purpose of this study was to investigate the presence and concentration of iron and zinc, in addition to MT-I/II expression, in the brains of aged cats. Through this, an attempt was made to study the correlation between MT-I/II, iron, and zinc with age-related pathological changes, aiming to obtain insights into the mechanisms of a normal aging feline brain. Finally, we prospected for evidence that might further support the hypothesis that cat could be a suitable natural animal model for aging and neurodegeneration.

## 2. Materials and Methods

### 2.1. Animals and Allocation into Groups

The study included thirty subjects. The brains from 3 young cats (1 to 2 years old) and 27 aged cats (7 to 20 years old) were obtained. The cats either died by natural causes or traumatic injury or were euthanized in the Companion Animal Clinic (Table 1). No history of neurological signs, mental or behavioral dysfunction was referred upon veterinary examination. Conventional necropsy was performed on all cats.

Vogt et al., in 2010, mentioned clustering of cats according to their age in the following four groups: Junior (7 months to 2 years old), Mature (7 to 10 years old), Senior (11–14 years old), and Geriatric (≥15 years old) [32]. In the present study, the cats were separated as follows: 12 belonged to the Mature group, 7 to the Senior group, and 8 to the Geriatric group. As controls were used the brains of three Junior cats (1.5, 2, and 2 years old).

### 2.2. Histopathological Examination and Scoring System

Coronal sections were taken from the right cerebral and cerebellar hemisphere. Tissue samples including the frontal and temporal grey (GM) and white (WM) matter, hippocampus, striatum, thalamus, and cerebellum were fixed in 10% neutral-buffered formalin, which was replaced after 2 days of fixation. Subsequently, they were embedded in paraffin wax and cut into 4–6 μm thick sections. Serial sections from each sample were stained with hematoxylin and eosin (H-E), Klüver–Barrera (KB), and Periodic acid–Schiff (PAS).

Histopathological lesions were scored using an optical microscope Nikon eclipse 50i. Lesions such as neuronophagia, satellitosis, chromatolysis, neuron and microglial lipofuscin deposits, neuronal vacuolation, neuronal necrosis and loss, perivascular microglia, neuroaxonal degeneration, white and grey matter vacuolation, cortical vascular fibrosis and hyalinosis, leptomeningeal fibrosis, leptomeningeal vessel (LV) fibrosis, LV and choroid plexus vessel (CPV) hyalinosis, LV calcification, choroid plexus (CP) epithelial and vascular fibrosis, and hemorrhages were scored as follows: **0**, no staining; **1**, detected in <25% of the examined areas; **2**, detected in 25–50% of the examined areas; **3**, detected in >50% of the examined areas. Additionally, the presence of axonal spheroids, Lafora-like bodies, and H-E-positive bodies was scored as follows: **0**, no staining; **1,** <10 bodies; **2**, 10–20 bodies; **3**, >20 bodies.

### 2.3. Prussian Blue–Diaminobenzidine (DAB)-Enhanced Histochemical Staining (Perl’s/DAB) and Scoring System

After deparaffinization and rehydration, sequential sections taken from the above samples were treated with 3,6% hydrogen peroxide for 15 min to block endogenous peroxidase activity. After washing in distilled water for 3–5 min, sections were incubated in a 1:1 freshly prepared mixture of 2% HCI and 2% potassium ferrocyanide at room temperature for 30 min. This protocol is consistent with the standard Perl’s reaction and gives blue coloration of the non-heme iron accumulation. Iron signals were additionally enhanced in a 3,3′-diaminobenzidine tetrahydrochloride (DAB) intensification procedure following Meguro’s modification [33], as described by Sukhorukova et al., 2013 [34]. Following the standard Perls’ reaction, the sections were washed 3 times for 5 min in DW (3 portions for 3–5 min) and then treated with a chromogen, DAB (Dako, Glostrup, Denmark), for 10–12 min. The exact time of treatment was adjusted according to development of optimal staining with DAB in the control preparations knowingly containing Fe^3+^. Finally, preparations were counterstained with 0.5% nuclear fast red (Sigma-Aldrich, Burlington, MA, USA).

Scoring of cytoplasmic neuronal and glial iron was evaluated by counting the number of positively stained cells in 10 consecutive high-power fields (40×). No staining was scored as **0**; a mean number of cells <20 as **1**; 20–30 cells as **2**; and >30 as **3** in the temporal and frontal GM and WM, hippocampus, striatum, thalamus, cerebellar cortex, and dentate nucleus.

In order to assess iron plaque load, we counted the total number of plaques per section. No staining was scored as **0**; PC <20 was scored as **1**; 20–30 as **2**; and >30 as **3** in the frontal and temporal GM and WM, hippocampus, striatum, thalamus, cerebellar GM and WM, and dentate nucleus.

Finally, detection of myelin-associated iron in the cerebral and cerebellar white matter and in the myelin-rich cerebral cortical layers, IV and V, was scored as follows: **0**, no staining; **1**, detected in <25% of the examined areas; **2**, detected in 25–50% of the examined areas; **3**, detected in >50% of the examined areas.

### 2.4. Immunohistochemistry Using Antibodies against MT-I/II, GFAP, and Aβ42

Immunohistochemistry was performed using a Super Sensitive Polymer-HRP IHC Detection System (BioGenex, Fremont, CA, USA). The primary antibodies used are listed in Table 2. Deparaffinized sections after EDTA microwave incubation (EnvisionFLEX, Target retrieval solution, high pH, Dako, Glostrup, Denmark) for 20–30 min at 500 watts (95–100 °C) for epitope retrieval were then treated with 3.6% hydrogen peroxide (H_2_O_2_)—methanol at room temperature for 30 min. They were subsequently incubated in 10% NGS at 37 °C for 30 min to avoid nonspecific reactions. The sections were then incubated at 4 °C overnight with one of the primary antibodies. After washing three times in PBS, sections were incubated with poly-HRP reagent (from Super Sensitive Polymer-HRP IHC Detection system) at room temperature for 40 min. Then, sections were washed with PBS and visualized with 0.05% 3-3′diaminobenzidine and 0.03% H_2_O_2_ in DW. Counterstaining was performed with Delafield hematoxylin. As positive controls for antibodies against MT-I/II, tissue sections from feline kidney samples were used, and for GFAP and Aβ42, sections from previously tested aged feline brain samples were used [6]. As negative controls, sections where the primary antibody was replaced by PBS were used.

### 2.5. Immunohistochemical Evaluation

MT-I/II expression was scored by calculating the mean number of positively stained neurons and astrocytes in 10 consecutive high-power fields (40× original magnification). The regions examined involved the grey and white matter of frontal and temporal GM and WM and the GM and WM (alveus and fimbria) of hippocampus, striatum, and thalamus. No staining was scored as **0**; a mean number of cells < 12 as **1**; 12–24 cells as **2**; and >24 as **3**.

For MT-I/II scoring in LV, GM and WM, blood vessels of the temporal and frontal lobes, hippocampal, thalamic and striatum blood vessels, frontal leptomeningeal cells (LCs), CPV, and CP epithelial cells, as well as ependymal cells, no staining was scored as **0** and positive staining was scored as **1**.

Tissue sections examined immunohistochemically for GFAP were evaluated by performing both morphological and quantitative analysis. Astrocytic GFAP grading based on morphological and quantitative criteria was classified according to Boos et al. (2021), as described in Table 3 [35]. The number of GFAP-positive astrocytes was assessed by calculating the mean number of astrocytes counted in five random fields (20× original magnification). 

We immunohistochemically evaluated the presence of Aβ deposits in the LV wall (cerebral amyloid angiopathy/CAA) and GM and WM CAA, Aβ accumulation in neurons and astrocyte Aβ deposition in CPV and CP epithelial cells, and Aβ plaque load.

For leptomeningeal CAA, we used the following scoring system: **0**, no staining; **1**, mild (Aβ deposits were detected in <25% of the vessels); **2**, moderate (Aβ deposits were detected in 25–50% of the vessels); **3**, severe (Aβ deposits were detected >50% of the vessels), in the leptomeninges of the frontal and temporal lobes as well as the cerebellum.

To score CAA (affecting arterioles and capillaries) in GM and WM of the temporal and frontal lobes as well as the cerebellum, areas in which more than 10 Aβ-positive vessels were detected in 20× original magnification were defined as “strongly CAA-affected areas”. CAA was classified as follows: **0**, no Aβ deposition in blood vessels; **1**, mild (less than 10 Aβ-positive blood vessels or only 1 strongly CAA-affected area was seen in the entire GM and WM); **2**, moderate (2 to 4 strongly CAA-affected areas were seen in the entire GM and WM); **3**, severe (more than 5 strongly CAA-affected areas were seen in the entire GM and WM) (modified by [36]).

Aβ accumulation in neurons and astrocytes was assessed by counting the total number of positively stained cells in 10 consecutive high-power fields (40×). Tissue sections from the temporal and frontal GM and WM, hippocampus, striatum, thalamus, cerebellar GM and WM, and dentate nucleus were evaluated. No staining was scored as **0**; a mean number of cells < 10 as **1**; 10–20 cells as **2**; and >20 as **3**. 

Aβ deposition in CPV and CP epithelial cells was evaluated and scored as follows: 0, no staining; **1**, mild (Aβ deposits were detected in <25% of the CPV and CP epithelial cells); **2**, moderate (Aβ deposits were detected in 25–50% of the CPV and CP epithelial cells); **3**, severe (Aβ deposits were detected >50% of the CPV and CP epithelial cells).

Finally, regarding Aβ plaque load, the total number of senile plaques (SPs) was counted in the frontal and temporal GM and WM, hippocampus, striatum, thalamus, cerebellar grey and white matter, and dentate nucleus. No staining was scored as **0**; a PC < 10 was scored as **1**; 10–20 as **2**; and > 20 as **3**.

### 2.6. Determination of Iron and Zinc in Brain Tissue by Inductively Coupled Plasma Mass Spectrometry (ICP-MS)

#### 2.6.1. Sample Preparation

Since trace amounts of analyte metals are ubiquitous in the environment and may be present in dust particles, care was taken to avoid external contamination during each step of the sample preparation. Efforts were made to use only polypropylene, polystyrene, and stainless-steel labware to minimize elemental backgrounds. Porcelain evaporation capsules were also used in the drying chamber in order to reach higher temperatures. Only powder-free gloves were used. Deionized and ultrapure water (Millipore Milli-Q water purification system) was used throughout the process. All sample storage containers and other equipment used were soaked for at least 24 h in 10% HNO_3_ and then rinsed copiously with ultrapure water.

Coronal sections from the left hemisphere of the frontal, parietal, occipital, and temporal lobes, as well as the left cerebellar hemisphere, were collected and then placed as a total in small polypropylene containers. The tissue segments were of the same width per region and were cut using an acid-washed quartz knife on an acid-washed polypropylene board. The sampled, complete tissue mass was mixed with 10 drops of ultrapure water and homogenized using the Ultra-Turrax Model T-25 homogenizer (Jahnke & Kunkel IKA, Staufen, Germany). Samples were dried at 105 °C for 24 h in a drying and heating chamber (Binder, Model ED56). The weight of the brain samples was determined before and after drying for at least 24 h until constant weight was achieved. The dry weight of each sample was 0.5–1.0 g. Three samples (Controls) were prepared as duplicates.

Samples were digested in sealed TFM vessels in a microwave digestion oven (Milestone MLS 1200) after 7 mL of 65% HNO_3_ (Merck, Suprapur, Darmstadt, Germany) and 3 mL H_2_O_2_ 30% *w*/*w* (Carlo Herba, Cornaredo, Italy) were added. After acid digestion, no visual solid residual remained. The Teflon vessels were allowed to cool to room temperature. Subsequently, samples were quantitatively recovered by filtration in 50 mL class A volumetric flasks and diluted to 50 mL with HNO_3_ 2%.

#### 2.6.2. Instrumentation 

An Agilent 7900 ICP-MS automated quadrupole ICP-MS instrument was used for the analysis of brain tissues for iron and zinc. The estimated concentrations of iron and zinc were expressed in μg/g.

### 2.7. Statistical Analysis

Iron and zinc concentrations in the brain of the selected cats were analyzed with one-way ANOVA using SPSS 28.0 (BM SPSS Statistics for Windows, Version 28.0. Armonk, NY, USA: IBM Corp.), while post hoc comparisons between groups were investigated using Duncan’s test. Statistical analysis of the scores (H-E, PAS, KB, and Perl’s/DAB histochemical staining, as well as Aβ, GFAP, and MT-I/II immunolabeling) was performed using chi-squared tests applied with SPSS as well as with the use of GraphPad Prism (version 9.1.2 for Windows^®^, GraphPad Software, San Diego, CA, USA). The level of significance was set at *p* < 0.05, and statistically significant trends (#) were underlined for *p* ≤ 0.1.

## 3. Results

Following natural death or euthanasia, a total of 30 cat brains were collected at necropsy, and their ages ranged from 1.5 to 20 years old. Almost a third of the cats were males (36.7%, n = 11), whereas two-thirds (63.3%, n = 19) were females. 

### 3.1. Histopathological Findings

Age-related lesions were not observed in the Controls. On the other hand, changes such as neuronophagia, satellitosis, chromatolysis, neuron and microglial lipofuscin deposits, neuronal necrosis and loss, perivascular infiltration of macrophages, neuroaxonal degeneration, and white and grey matter vacuolation were seen to a variable degree in the frontal and temporal lobe as well as the cerebellum of aged cats (Table 4, Appendix A).

Neuronophagia and satellitosis were distributed in layers II-VI in the cerebral cortices, primarily in the frontal lobe, the pyramidal layer of Cornu Ammonis (CA) of the hippocampus, the Purkinje cell layer, and the dentate nuclei. A constant lesion observed in all elderly cats was chromatolysis, particularly in Betz cells, the pyramidal cell layer of the hippocampus, and the Purkinje cell layer. High severity of neuronal loss was noticed in cats aged 20 years old. Complete Purkinje cell loss in some regions was observed in 13 cats over 10 years old (case Nos 4, 16, 17, 19, 20, 21, 23, 24, and 26–30). Additionally, in five of these cases, there was also a reduction in granule cell density (case Nos 20, 23, 24, 27, and 29). Neuronal vacuolation was noticed in 9 of the 27 aged cats. Single or multiple variable size intracytoplasmic vacuoles were primarily found in the neurons of the cerebral cortices as well as in Purkinje and Golgi type II neurons.

Regarding vascular lesions, medial hyalinosis and adventitial fibrosis of small arteries and arterioles were found in the GM and WM, meninges, and CP. Additionally, calcification of the LV wall was noted in seven aged cats over 8 years old. Leptomeningeal thickening due to an increase in collagen fibers was observed in almost all brains (25/27 cases).

Three types of bodies were observed. Lafora-like bodies detected in the aged brains were more numerous in the cerebellum than in the temporal and frontal lobes. They were distributed within all cortical and cerebellar layers and white matter as well as the dentate nucleus, and they were more prominent in the grey than the white matter. Spheroids were displayed both in grey and white matter in five cases (case Nos 6, 16, 24, 29, and 30). In the cerebellum, they were found in the white matter in three cases (case Nos 4, 11, and 24) and in the granular cell layer in one case (case No 26). H-E-positive bodies were noticed in 14 cats over 8 years old. These homogeneous, round-to-ovoid, non-membrane-bound basophilic bodies were distributed in the neuropil in layers I-II of the cerebral cortices, in the cerebellar molecular layer, and dentate nuclei. The above formations were not associated with reactive lesions in the adjacent neuropil. A few aged animals (six cases) had small foci of hemorrhages (case Nos 9, 11, 12, 24, 26, and 30).

### 3.2. Perl’s/DAB Scoring

Perl’s/DAB-stained iron deposits were observed within neurons, glial cells, and their cytoplasmic projections in the grey and white matter of the temporal lobe in 25/30 cats. Similar deposits were found in the hippocampus (29/30), thalamus (25/30), striatum (15/30), frontal lobe (26/30), and cerebellum (29/30). Positively stained cells were noticed in all cerebral cortical layers (I–VI). The hippocampus revealed positive neuronal staining, most pronounced in the pyramidal cell layer of the CA and dentate gyrus. In the cerebellum, Purkinje cells were primarily stained, but a small number of basket and Golgi type II cells were also positive. Perl’s/DAB staining was predominantly detected within the nucleoli, while nuclear and perikarial staining of varying degrees was also observed.

Perl’s/DAB histochemical staining revealed the presence of extracellular deposits in the gray and white matter forming two types of iron plaques (IPs) that shared morphological similarities with Abeta-positive senile plaques. Thus, since the first type was characterized by focal spherical deposits with a dense core, it was named “condensed iron plaque”. The second type was characterized by large, diffuse, and poorly delimited deposits and was named “diffuse iron plaque”(Appendix A). Condensed IPs were the predominant type, and both types were detected in all examined brain regions. Plaque staining in the cortex was present in all cortical layers. In animals with scanty deposits, the plaques were distributed in IV/V cortical layers. The IPs were noticed in greater numbers in grey matter compared to white matter. In the hippocampus, IPs were principally noticed in the CA2, CA3 subfields, stratum oriens, and stratum lacunosum-moleculare of the CA, as well as the dentate hilus. In the cerebellum, the abundance of IPs was present in the molecular layer. Plaque formation was also noticed in the striatum and thalamus (Figure 1).

In addition to IPs, band-like iron deposition was found in the white matter of the frontal and temporal lobes and the cerebellum, representing myelin-associated iron. Band-like depositions were also noticed in the thalamus in 4/30 cats (9 years old) and in the fimbria of the hippocampus in 3/30 aged cats (>12 years old) (Table 5, Appendix A).

### 3.3. Immunohistochemical Results

MT-I/II immunoreactive cells were detected in the grey and white matter of the temporal and frontal lobes, hippocampus, thalamus, and striatum, as well the internal capsule of all 30 animals. They were seen throughout cerebral cortical layers II to V. In the hippocampus, the highest MT-I/II immunolabeling was detected in the CA4 and CA3 subfields, stratum oriens, and stratum lacunosum-moleculare of CA, as well as the dentate molecular layer and hilus. The presence of MT-I/II-positive astrocytes was a constant finding in all cat brains examined, contrarily to neurons, which were occasionally positively stained. A small number of immunoreactive neurons was noticed only in 1 aged cat in the temporal cortex (case No 28), whereas positive staining of the frontal cortex was observed in 14 cases (case Nos 4–6, 9, 11, 14, 16–18, 23–26, and 28). Noteworthily, the Controls did not display neuronal immunoreactivity. Detailed results are summarized in Table 6 and Appendix A.

MT-I/II immunoreactivity was shown in both the nucleus and cytoplasm of astrocytes and in the cytoplasm of neurons and was more intense in cats with severe age-related histopathological changes. A small number of MT-I/II immunoreactive astrocytes was observed in the brain of the control cats. Animals’ brains showed positive reactions in the leptomeningeal, CP, and ependymal cells; LV and blood vessels in GM and WM of both lobes; hippocampal, thalamic, striatal, and CPV regions. Immunoreactive LCs were detected only in the frontal lobe in 11 out of 30 cats, involving only elderly animals. CP epithelial cell immunolabeling was detected in 20 out of 30 cats. MT-I/II immunopositive astrocytes were occasionally observed around blood vessels in cerebral GM and WM (Figure 2). 

Frontal GM: Controls vs. Geriatric group: *p* = 0.024, Mature vs. Geriatric group: *p* = 0.002, Senior vs. Geriatric group *p* = 0.095; Frontal WM: Controls vs. Geriatric group: *p* = 0.006, Mature vs. Senior group: *p* = 0.056 ^#^, Mature vs. Geriatric group: *p* < 0.001; Frontal meningeal cells: Controls vs. Geriatric group: *p* = 0.008, Mature vs. Geriatric group: *p* = 0.002, Senior vs. Geriatric group: *p* = 0.020; CP: Controls vs. Geriatric group: *p* = 0.040, Controls vs. Senior group: *p* = 0.044, Mature vs. Senior group: *p* = 0.003, Mature vs. Geriatric group: *p* = 0.002; CPV: Mature vs. Controls: *p* = 0.089 ^#^, Mature vs. Senior Group: *p* = 0.021, Mature vs. Geriatric Group: *p* = 0.016; Hippocampal vessels: Controls vs. Geriatric group: *p* = 0.024, Mature vs. Geriatric group: *p* = 0.009; Thalamic vessels: Mature vs. Senior Group: *p* = 0.024, Mature vs. Geriatric Group: *p* = 0.012; Striatal vessels: Mature vs. Senior Group: *p* = 0.023, Mature vs. Geriatric Group: *p* = 0.012.

GFAP immunostaining showed diffuse astrogliosis in the frontal and temporal lobes, glia limitans, hippocampus, and cerebellum. Detailed results are summarized in Table 7 and Appendix A. In the control cats, there were a few GFAP immunoreactive astrocytes with long and thin processes, while the nucleus remained unstained. Astrocytes were labeled dark brown with the anti-GFAP primary antibody and were generally distributed throughout all cortical layers (Figure 3).

GFAP-immunopositive astrocytes were observed in layers I-VI in 22 cases, in layers I-II in 3 cases (case Nos 9, 13, and 14), and throughout layers I-IV in 2 cases (case Nos 7 and 10) in the cerebral cortices. Increased GFAP immunoreactivity was primarily noticed in the hilar astrocytes of the dentate gyrus of the hippocampus and in all cerebellar cortical layers and white matter. Control cats had grade 0, and all 27 aged cats presented all three grades on combined evaluation. Both the morphological alterations and the cell density were more severe in the Geriatric group (most of the animals were evaluated as grade 3 for the majority of the variables).

**Aβ deposition** scoring in the examined regions of all aged animals is illustrated in Table 8 and Appendix A. Domestic cats at an age above 7 years old showed Aβ pathology. None of the cat brains included in the Controls was positively stained for Aβ. Immunostaining for Aβ peptide revealed extracellular deposits forming two types of senile plaques (SPs) in the extracellular space: diffuse granular, which was the most common type observed, and stellate, which was rarely detected. The diffuse type was further divided into two subtypes: condensed and “cloud-like” (Figure 4).

The condensed subtype was characterized by well-circumscribed dense spherical accumulation of Aβ-positive antigenic material. Morphologically, they had an immunoreactive core and a less well-outlined crown. Degenerating and/or necrotic neurons, glial cells, and blood vessels were occasionally observed in the center of condensed plaques.

On the contrary, “cloud-like” plaques had sparse homogeneous distribution, an irregular to round shape, and granular texture. They were ill defined and variably sized: from small to extremely large deposits.

Finally, stellate plaques were characterized by a dense center from which fibrils of Aβ42 were radiating to the periphery. Radiating fibrils had lengths and thicknesses of varying sizes, giving a star-like appearance.

SPs were found in 21/27 aged cats over 7 years old in gray matter, and in 12 of them, there were also deposits in white matter. They were distributed within layers III to VI of the cerebral cortex. Plaque formation was also observed in 10 cats over 9 years old in the hippocampus, in 7 cases over 12 years old in the thalamus, and in 9 cats over 12 years old in the striatum. No plaque formation was detected in the sections of cerebellum.

Diffuse plaques of both subtypes were observed in all 21 aged cats, which revealed plaque formation, except in case No 15, where there was only one condensed plaque located in the VI cortical layer of the temporal lobe. Stellate plaques were found in eight cats over 14 years old (case Nos 21, 23–25, and 27–30). In particular, cats of the Geriatric group (15 to 20 years old) had higher scores regarding extracellular deposits. The severity of pathology and the number of cats with SPs increased with age (frontal and temporal GM and WM SPs: *p* < 0.001; thalamic SPs: *p* = 0.092 ^#^; and striatal SPs: *p* = 0.047).

In addition to plaques, Aβ-positive staining forming band-like deposition was also found in cats over 12 years old. They were distributed within the hippocampal dentate molecular layer and stratum lacunosum-moleculare (case Nos 27, 29, and 30); hippocampal alveus, fimbria, and stratum oriens (case No 28); temporal cortical layer V to VI (case Nos 18, 19, 26, 29, and 30); and both in the frontal and temporal cortex in layers V–VI (case Nos 20, 21, 23–25, 27, and 28).

Aβ immunoreactivity was also detected in the wall of the LV, CPV, cerebral and cerebellar vessels, and around the wall of cerebral capillaries. Intracytoplasmic accumulation of Aβ was detected in the neurons of the cerebral cortex, hippocampus, thalamus, striatum, and cerebellum. Increased Aβ42 immunostaining was noticed within layers III to VI of the cerebral cortex, in the pyramidal layer of the CA, and Purkinje and Golgi type II cells of the cerebellar cortex. Additionally, CP epithelial cells in all elderly cats showed Aβ42 immunoreactivity.

Besides neuronal accumulation, astrocytes also showed intracytoplasmic immunolabeling of Aβ. This was observed in the frontal and temporal lobes and the cerebellum in six aged cats (case Νos 5, 12, 21, 24, 26, and 28).

### 3.4. ICP-MS Metal Analysis

The levels of iron and zinc in the analyzed brain tissues are reported in Table 9. The highest iron and zinc levels were detected in case Nos 21 and 26, while the lowest was detected in case No 1. For both metals, no statistical significance was noticed among the age groups (*p* > 0.1). However, increasing age was associated with a mild increase in iron levels in the brain, as illustrated in Figure 5. Regarding zinc, a mild numerical increase in the Mature and Senior groups was displayed compared to the Controls (Figure 5). The Geriatric group concentration of zinc was similar to the Controls.

## 4. Discussion

An aging brain is strongly associated with several functional and morphological alterations. Brain biometal dyshomeostasis, such as iron and zinc, contributes to increased oxidative stress, which is related to severe neuronal damage in normal aging and neurodegenerative diseases in both humans and animals [26,37]. In the present study, we investigated the association of age-related histopathology, including Aβ deposition, with brain MT-I/II and metal ion accumulation.

Most of the histopathological alterations demonstrated using H-E appeared to increase with age (*p* < 0.1), as has been described in previous reports on aged cats [4,6,7,27]. Neuronophagia, satellitosis, and chromatolysis were more severe in animals over 17 years old. Neuronal necrosis and loss observed in the cerebellum mostly affected Purkinje neurons and then Golgi type II cells, dentate nucleus, and granule cells. Interestingly, regional complete Purkinje cell loss was observed and sometimes alongside reduced granule cell density (case Nos 20, 23, 24, 27, and 29). This finding is in agreement with Brellou (2006) who further noticed reduced Golgi type II cells [6]. Furthermore, Zhang et al. (2006) observed a severe loss of neurons in the cerebellar molecular layer of aged cats [27]. In addition, axons exhibited degenerative changes characterized by myelin sheath rupture and segmental loss and the presence of vesicular formations. The latter observation was more prominent in the Geriatric group. Another observation worth mentioning, which has also been mentioned by Brellou (2006), is that while the number of Lafora-like bodies was greatly increased in aged cats over 11 years old, they showed decrement in size with age [6].

In addition to the age-related changes detected by several authors, Brellou (2006) also observed spherical-to-ovoid structures stained exclusively with H-E staining in cerebral cortical gray and white matter and in the cerebellar molecular layer and dentate nuclei of aged cats [6]. Interestingly, we detected similar H-E-positive bodies in the cerebral GM and WM matter, the cerebellar molecular layer, and the dentate nuclei of 14 cats over 8 years old. In both cortices, the H-E-positive bodies were distributed within layers I and II. The above bodies were detected only with H-E staining, and an age-dependent increase was noted (*p* = 0.014). Further investigation could be undertaken to clarify the origin and role of these structures in the aged feline CNS.

Regarding brain iron accumulation, we performed the Perl’s/DAB staining method, a modified Perl’s staining described by Meguro et al. (2007) [33]. The Meguro modification primarily stains Fe^3+^, but also Fe^2+^, and is widely used in order to achieve higher sensitivity [33]. DAB-enhanced Perl’s methods have been applied in the brain of aged dogs [38], in the brain of aged rats [39,40], in human brain tissue with AD pathology [41,42,43], in the brain of aged wild-type mice [44], and in a mouse model (APP/PS1) of AD [45].

Previously, researchers have observed the extracellular deposition of iron and, particularly, the formation of plaques stained with Perl’s/DAB in the cortex, hippocampus, cerebellum, caudoputamen, and thalamic nuclei of mice and in the frontal cortex of AD patients [41,43,45]. Moreover, Kim E. et al. (2021) noticed the presence of iron-containing Aβ plaques throughout the cerebral cortex, hippocampus, and thalamus of rodents [46]. According to Sands et al. (2016), the IPS demonstrated either dense core staining or sometimes diffuse. This is in line with our observations in cats regarding both condensed and diffuse IPs in the same regions [45].

Along with plaques, prior research has indicated iron-myelin-associated staining in the cerebellar white matter of rats; in the globus pallidus, ventral pallidum, basal ganglia, and substantia nigra of mice; in the globus pallidus of aged dogs; and in the frontal white matter and myelin-rich cortical layers IV and V of human AD patients [38,40,41,43,45,47]. Similarly, in our study, iron–myelin-associated staining was also observed for the first time in the aging feline brain located in the frontal, temporal, and cerebellar white matter; thalamus; and hippocampal fimbria.

Sands et al. (2016) also noticed punctate staining in neurons located in the cortex, CA hippocampal layer, and dentate gyrus [45]. Neurons occasionally revealed nuclear, nucleolar, and/or cytoplasmic staining, without describing, in detail, the exact cellular distribution. Meguro et al. (2008) also demonstrated cytoplasmic and nuclear neuronal staining in the cerebellar nuclei and cortex, globus pallidus, and in many brain stem structures in a rat brain [40]. Similarly, our results showed nucleolar, nuclear, and/or cytoplasmic staining within neurons delocated in the cerebral cortices, the hippocampal stratum pyramidale layer, the dentate gyrus, the thalamus, and the cerebellum. However, in our study, we demonstrated that the nucleolus was the primary iron-labeled site in the aged feline brain. Previous studies have also demonstrated that the neuronal nucleolus is a “hot spot” for iron, showing a higher concentration than the nucleus and the perikaryon in rat brains [48,49].

It has been suggested that aging might result in an increased concentration of total iron and lead to the accumulation of iron in neurons and glial cells in humans [14]. Particularly, glial cells also exhibit Perl’s/DAB staining. Earlier studies have shown iron-positive oligodendrocytes, astroglia, and microglia in aged rats, mice, dogs, and humans [38,40,41,42,43,45]. In our study, no double staining was performed to demonstrate simultaneous glial cell type and iron accumulation. However, based on cell morphology, including the detection of long cytoplasmic processes, these cells were identified either as astrocytes or as activated microglia.

It has been widely acknowledged that essential metal dyshomeostasis is strongly correlated with aging [11,13,14,16,17,26]. Increased iron concentrations have been frequently associated with age in the brain of humans, rats, mice, O. degus, and dogs [11,14,38,39,40,50,51]. In the present study, a statistically significant age-dependent increment in iron accumulation was demonstrated using Perl’s/DAB staining, exclusively regarding plaque load in the frontal grey and white matter (*p* = 0.088 ^#^) and the thalamus (*p* = 0.025), as well as myelin-associated iron in the cerebellum (*p* = 0.002). However, our findings using ICP-MS did not show statistically significant differences regarding iron levels among the age groups (*p* > 0.1), and iron concentrations showed a minor age-related increase, as shown in Figure 5.

Concerning zinc levels, there is scientific controversy for the age-dependent increment in zinc concentration in the brain using ICP-MS analysis. While Takahashi et al. (2001) showed that zinc levels did not vary with age in rodents, Zatta et al. (2008) detected an age-related significant increase in zinc in the cerebellum and thalamus in cattle [26,52]. Previous reports have demonstrated a significant increment in zinc concentrations in the cortex and hippocampus of O. degus and in the globus pallidus of mice. On the contrary, zinc levels were not significantly increased with age in the hippocampus of mice [11,50]. Our ICP-MS results did not show statistically significant differences regarding zinc levels among the age groups (*p* > 0.1). However, higher levels of zinc were observed in the cats of the Mature and Senior group aged from 7 to 14 years old compared to those of the Controls and Geriatric groups.

Metal concentration analysis with ICP-MS was not determined regionally but in a total of five regions. As a result, it was not feasible to examine the precise distribution and calculate the concentration of these essential metals in each specific region separately in the feline brain. But, as regards iron deposits in situ in several areas of the feline brain, the detailed distribution, morphology (types), and localization was achieved with Perl’s/DAB staining. The latter was proven to be a precious method, as for many other authors who studied human AD, mice, and canine brains [38,41,43,44,45].

Another constant finding in the present study worth mentioning was the intense expression of MT-I/II in astroglia, a metalloprotein involved in metal binding and protection against reactive oxygen species [20,22]. Although MT-I/II is primarily produced by astrocytes [53], other cell types also show MT-I/II expression [21,54]. Similarly, our immunohistochemical results showed predominantly MT-I/II immunoreactive astrocytes in addition to neurons, LCs, CP cells, and ependymal cells, as well as endothelial cells. Furthermore, MT-I/II expression was also demonstrated in vascular tunica media.

Earlier studies have also noticed MT-I/II immunolabeling in domestic animals. Shimada et al. (1998) demonstrated increased MT-I/II astrocytic immunostaining with age in the intact brain areas of 13 dogs with neurological signs [23]. Zatta et al. (2008) found MT-I/II immunoreactive astrocytes in the cerebellum, frontal and parietal cortex, and ependymal cells in the brain of both young and aged cattle. However, they did not note any significant difference in the distribution of MT in association with age [26]. Our findings are almost in accordance with those of Kojima et al. (1999), who found MT-I/II immunopositive astrocytes, both in the cytoplasm and nucleus, ependymal cells, and CP epithelial cells in the brain of aged dogs [24], a finding also observed in monkeys [55], rats [56,57], and mice [56,58]. Positive astrocytes were distributed in the cerebral cortical layers from III to V, white matter, hippocampus, thalamus, periventricular area, and around the blood vessels of each region [24]. Morita et al. (2005) also demonstrated the expression of MT-I/II, mainly in astrocytes around the blood vessels of aged dogs [38]. The latter observation regarding the perivascular presence of astrocytes combined with the fact that astrocytes are an essential part of the blood–brain barrier and that they ensheath blood vessels indicates that MT-I/II plays a pivotal role, acting as a barrier for dysregulated metal uptake.

We further observed MT-I/II immunoreactivity in neurons of all regions examined. Specifically, neuronal MT-I/II expression has scarcely been demonstrated in reports on humans, mice, rats, and sheep [20,21]. In our study, MT-I/II-positive staining was demonstrated in a noticeable number of cats (14 cases). This finding in the feline brain might be attributed to cytoplasmic MT-I/II extracellular release and then its internalization by neurons, eventually promoting neurite outgrowth and neuronal survival (as reviewed in [21]).

In conclusion, statistical analysis revealed a significant increment in MT-I/II expression. All the variables examined, except MT-I/II immunoreactivity in the temporal and frontal leptomeningeal and GM and WM blood vessels, were statistically significant (*p* < 0.1). Taking into consideration the fact that we compared different age groups, it might be concluded that MT-I/II expression is positively correlated with age in the feline brain. Our results agree with Mocchegiani et al. (2001), who also observed increased MT-I/II immunoreactivity in the brain of old rats and proposed MT-I/II as a potential marker of aging [25].

In the feline brain, a correlation between amyloid beta deposition and increasing age has been widely documented [59]. In the current study, both intracellular and extracellular Aβ deposits were observed. The latter were detected from the age of 7 years old. Previous studies of feline Aβ pathology have reported deposits in cats over 7.5 years old [6,60,61,62], but in a recent study, the youngest cat was 4 years old [8]. However, additional data regarding the animal’s detailed history were not mentioned.

Interestingly, among the brain regions we examined, the cerebellum showed the lowest load of Aβ, and extracellular deposits were absent. Cerebellar Aβ deposits have been reported in aged cats [6,8]. According to Thal et al. (2002), the cerebellum is involved in end-stage AD [63]. In the present study, it is likely that our cases might have developed cerebellum deposition if they had lived longer [8].

Although neurons were assumed for a long time as the exclusive cell type capable of producing Aβ, additional studies have indicated that astrocytes also generate beta amyloid [64,65,66,67,68,69]. Both cell types express BACE1 (β-site APP cleaving enzyme 1), which is indispensable for Aβ production, and its expression can be increased due to cellular stress [30,68]. Age-dependent astrogliosis has been reported in animals including dogs [23,70], cattle [71], horses [72], and cats [6,28]. Accordingly, in the present study, the identification of GFAP-positive astrocytes and further classification into four separate grades were performed based on Boos et al. (2021), and a statistically significant age-related increase in size and number of positive astrocytes was observed [35]. Given that astrocytes outnumber neurons in the brain and reactive astrogliosis is associated with aging, these could lead to astrocyte contribution in Aβ production in aged humans and/or animals [30,68]. In our study, we documented, for the first time, the intracytoplasmic immunohistochemical detection of Aβ in astrocytes in the aged feline brain.

Aβ extracellular deposits have also been related to high levels of iron and zinc ions in AD [14,15,29,43,46]. Previous research has demonstrated a positive correlation between iron accumulation and Aβ plaques in the frontal cortex of humans with AD and throughout the brain of rodents [43,46]. Increased total brain iron has been associated with early Aβ plaque formation in a mouse model of AD [73]. Furthermore, studies have shown that MT-I/II is expressed in high levels in AD and is also associated with Aβ plaques in the hippocampus of AD animal models [21]. In the current study, two cases (case Nos 21 and 26; 14 and 17 years old, respectively) showed the highest brain concentration of both iron and zinc among the animals when analyzed with ICPMS. Interestingly, those cats also had a very high Aβ plaque load in the temporal and frontal lobes. Particularly, case No 21 had 125 plaques in the frontal lobe and 112 SPs in the temporal lobe, while in case No 26, 62 and 99 SPs were found in the same regions, respectively. The same cats also displayed histochemically high iron plaque burden and high expression of MT-I/II. These data suggest that there might be a positive correlation between iron and zinc levels, as well as MT-I/II expression, with Aβ extracellular deposition in the brain of aged cats.

To the authors’ knowledge, this is the first study to investigate the feline brain for MT-I/II expression, iron and zinc ion concentrations and iron ion distribution as well. Also, an attempt was made to further compare these findings with age-related histopathology including Aβ deposits. In light of the evidence raised, we strongly suggest that cats merit further investigation as a natural animal model of brain aging and neurodegenerative diseases.

## Figures and Tables

**Figure 1 brainsci-13-01115-f001:**
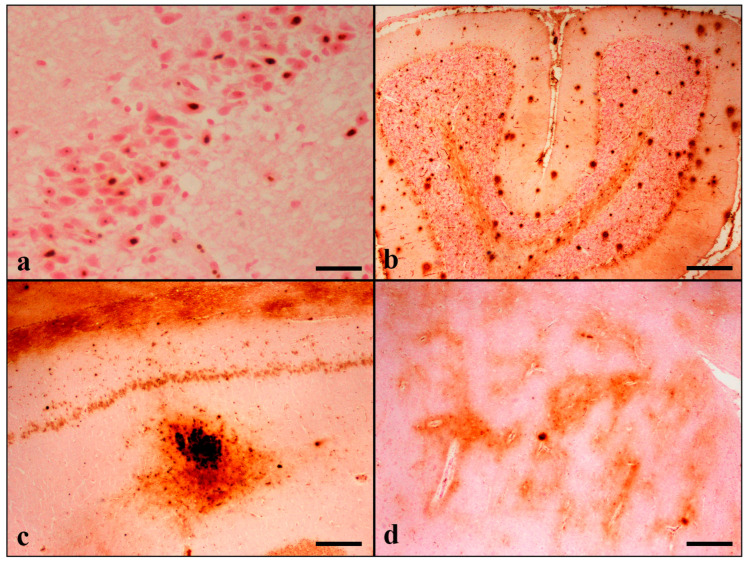
(**a**) Intense nuclear and nucleolar staining of neurons of the dentate gyrus (case no 5). (**b**) Condensed and diffuse IPS in all cerebellar layers of the cerebellar cortex and in the white matter. Purkinje, Golgi type II, basket, and glial cells are stained (case no 9). (**c**) Numerous coalescing condensed iron plaques throughout the hippocampal stratum lacunosum-moleculare and stratum radiatum. Neurons, glial cells, and axons around the coalescing plaques are also stained. CA1 pyramidal cell layer shows cytoplasmic staining. Notice the diffuse band-like iron deposits in the hippocampal alveus indicating myelin-associated iron (case No 27). (**d**) Multiple Perl’s/DAB-positive areas representing myelinated fibers as well as positive glial cells in the thalamus. Two condensed iron plaques are also displayed (case No 6). Perl’s/Dab. (**a**) Bar = 25 μm, (**b**–**d**) Bar = 250 μm.

**Figure 2 brainsci-13-01115-f002:**
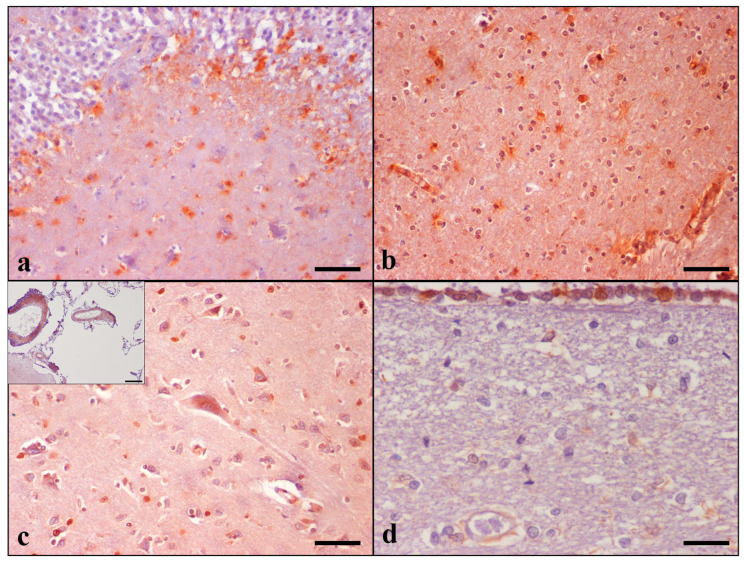
MT-I/II immunolabeling. (**a**) Positive MT-I/II immunostaining of astrocytes in the DG hilus and CA4 region of the hippocampus (case No 26) and (**b**) in the temporal cortex. Immunoreactive astrocytes around blood vessels (lower right) (case No 30). (**c**) Immunolabeling is also occasionally present in the neuronal cytoplasm of the temporal cortex (case No 28). **Inset:** intense immunostaining of the tunica media of the temporal LMV (Case No 24). (**d**) Ependymal cells showing positive MT-I/II staining in both cytoplasm and nuclei. Positively stained astrocytes are also displayed. IHC, DAB chromogen, hematoxylin counterstain. (**a**–**c**) Bar = 50 μm, ((**c**) inset) Bar = 100 μm, (**d**) Bar = 25 μm.

**Figure 3 brainsci-13-01115-f003:**
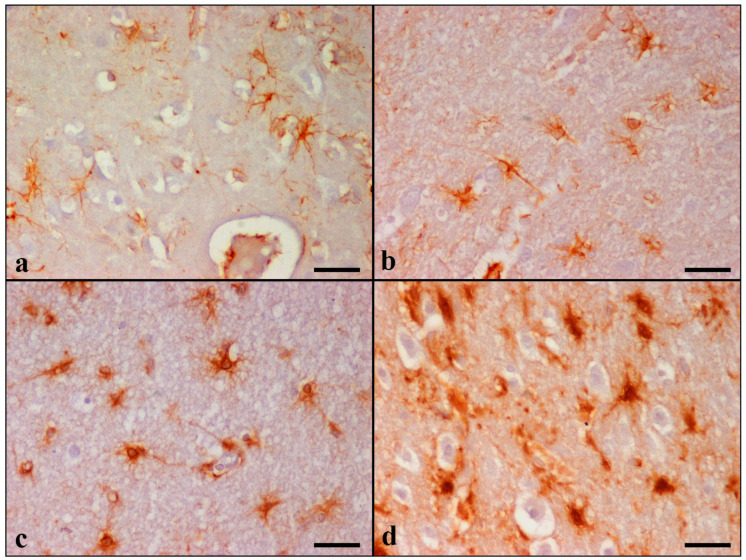
Tissue sections from temporal lobes of cats positively stained for GFAP. (**a**) Case No 2 aged 2 years old: grade 0 in temporal GM, (**b**) Case No 13 aged 9 years old: grade 1 in temporal WM, (**c**) Case No 19 aged 11 years old: grade 2 in temporal WM, (**d**) Case No 27 aged 17 years old: grade 3 in temporal GM. IHC, DAB chromogen, hematoxylin counterstain. (**a**–**d**) Bar = 25 μm.

**Figure 4 brainsci-13-01115-f004:**
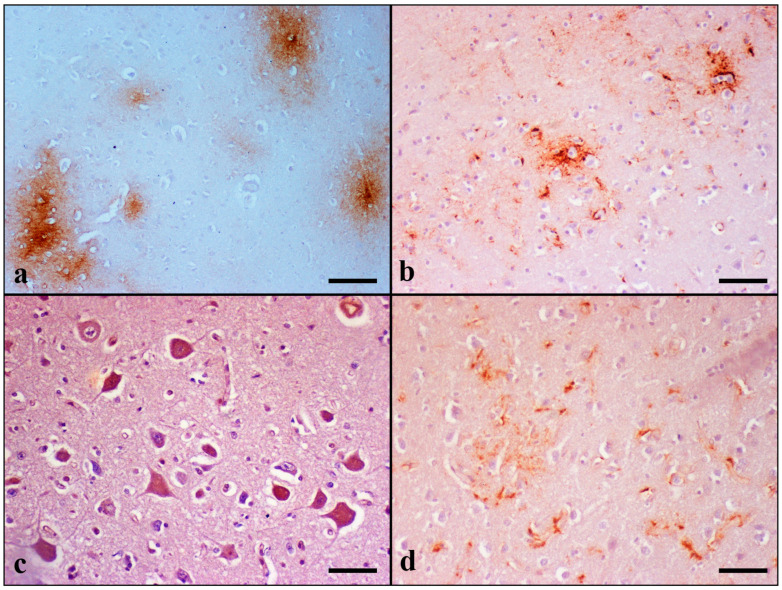
Aβ42 immunolabeling in the brain of aged cats. (**a**) Condensed and cloud-like amyloid plaques within the neuropil of the temporal cortex. Within condensed plaques can be observed intact non-stained neurons (case No 6). (**b**) Two Aβ42-positive stellate plaques can be observed in the neuropil of the temporal cortex. Glial cells with long and thin or short and thick cytoplasmic processes, probably reactive astrocytes, are also positively stained (case No 29). (**c**) Intense cytoplasmic Aβ42 immunostaining in Betz neurons (layer V) of the frontal cortex (case No 4). (**d**) Aβ42 is occasionally present in the cytoplasm and cytoplasmic processes of astrocytes within the frontal cortex. A cloud-like plaque surrounded by positively stained astrocytes is obvious (case No 24). IHC, DAB chromogen, hematoxylin counterstain. (**a**) Bar = 100 μm, (**b**–**d**) Bar = 50 μm.

**Figure 5 brainsci-13-01115-f005:**
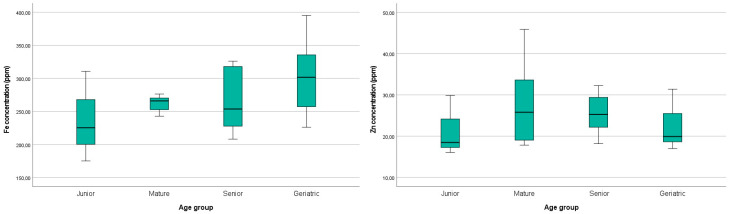
The results were not statistically significant (*p* > 0.1) for both iron and zinc. A mild age-dependent numerical increase in iron and mildly elevated zinc concentration in the Mature and Senior groups were noticed.

**Table 1 brainsci-13-01115-t001:** Age, gender, breed, pathological data, and reason for euthanasia or death of the aged cats.

	Case No.	Age (Years)	Gender	Breed	Reason for Euthanasia or Death		Case No.	Age (Years)	Gender	Breed	Reason for Euthanasia or Death
	1	1.5	♂	DSH	Gastric and hepatic neoplasia/hypovolemic shock—death		16	>10	♀	DSH	Accident—death
Controls	2	2	♂	DSH	Accident—death		17	12	♀	DSH	Septic shock—death
	3	2	♀	DSH	Accident—euthanasia		18	12	♀	DSH	Chronic renal and hepatic failure—death
	4	10	♂	DSH	Acute leukemia—euthanasia	Senior Group	19	11	♂	Siam	Triaditis—death
	5	7	♀	Persian	Accident—euthanasia		20	13	♂	DSH	Alimentary lymphoma—death
	6	7–10	♀	DSH	Accident—death		21	14	♂	DSH	Renal neoplasm—euthanasia
	7	7	♂	DSH	Chronic renal failure—death		22	>10	♀	DSH	Chronic renal failure—death
	8	7	♀	Persian	Arterial thromboembolism—death		23	20	♀	Persian	Polycystic kidney, liver, and pancreas disease—death
Mature group	9	9	♂	DSH	Hypertrophic cardiomyopathy—death		24	20	♀	Persian	Chronic renal failure—euthanasia
	10	9	♀	DSH	Status epilepticus—pulmonary—death		25	15	♂	DSH	Melanoma—euthanasia
	11	8	♀	DSH	Hypovolemic shock—death	Geriatric Group	26	17	♂	DSH	Accident—death
	12	8	♀	DSH	Lower urinary tract disease—death		27	17	♀	DSH	Chronic renal failure—euthanasia
	13	9	♀	DSH	Accident—death		28	16	♀	Persian	Chronic renal and hepatic failure—euthanasia
	14	10	♀	DSH	Chronic renal failure—euthanasia		29	15	♀	DSH	Chronic renal failure—death
	15	7	♀	Siam	Polycystic kidney disease—euthanasia		30	16	♂	DSH	Peritoneal mesothelioma—death

**Table 2 brainsci-13-01115-t002:** Antibodies used in the current study.

Antibody	Dilution	Source	Pretreatment
Anti-MT-I/II	1:200	Novus Biologicals	EDTA high pH (EnvisionFLEX, Target retrieval solution, high pH, Dako, Glostrup, Denmark)
Anti-GFAP	1:400	Dako, Denmark	None
Anti-amyloid β42	1:800	EMD Millipore, USA	Formic acid 87% (AnalaR NORMAPUR)

**Table 3 brainsci-13-01115-t003:** Combined evaluation of GFAP-positive astrocytes in the brain of aged cats, using both quantitative and morphological criteria (Boos et al. 2021).

	Grade 0	Grade 1	Grade 2	Grade 3
Quantitative criteria	0–30 cells	16–60 cells	40–100 cells	>77 cells
Morphological alterations	Absence of nucleus alterations and a few mildly stained cells with long, thin, well-ramified cytoplasmic processes	Mildly increased nuclei volume and mildly stained long, thin, well- ramified cytoplasmic processes	Moderately increased nuclei volume and moderately stained long, moderately thickened cytoplasmic processes	Severely increased nuclei volume (gemistocytes) and intensely stained thickened cytoplasmic processes trespassing other cells processes

**Table 4 brainsci-13-01115-t004:** Histopathological scoring of age-related brain lesions in mature, senior, and geriatric cats.

Groups						
Investigated Parameter	Grading Score	Controls	Mature	Senior	Geriatric	Significance (Chi Square)
Neuronophagia	0	100% (3)	0%	0%	0%	*p* < 0.001
	1	0%	91.7% (11)	71.4% (5)	0%	
	2	0%	8.3% (1)	28.6% (2)	50% (4)	
	3	0%	0%	0%	50% (4)	
	Total	100% (3)	100% (12)	100% (7)	100% (8)	
Satellitosis	0	100% (3)	0%	0%	0%	*p* < 0.001
	1	0%	75% (9)	42.9% (3)	0%	
	2	0%	16.7% (2)	57.1% (4)	50% (4)	
	3	0%	8.3% (1)	0%	50% (4)	
	Total	100% (3)	100% (12)	100% (7)	100% (8)	
Chromatolysis	0	100% (3)	0%	0%	0%	*p* < 0.001
	1	0%	66.7% (8)	14.3% (1)	0%	
	2	0%	33.3% (4)	85.7% (6)	50% (4)	
	3	0%	0%	0%	50% (4)	
	Total	100% (3)	100% (12)	100% (7)	100% (8)	
Neuronal lipofuscin deposits	0	100% (3)	0%	0%	0%	*p* < 0.001
	1	0%	41.7% (5)	14.2% (1)	0%	
	2	0%	58.3% (7)	42.9% (3)	12.5% (1)	
	3	0%	0	42.9%(3)	87.5% (7)	
	Total	100% (3)	100% (12)	100% (7)	100% (8)	
Neuronal vacuolation	0	100% (3)	66.7% (8)	85.7% (6)	50% (4)	Non-significant
	1	0%	33.3% (4)	14.3% (1)	25% (2)	
	2	0%	0%	0%	25% (2)	
	3	0%	0%	0%	0%	
	Total	100% (3)	100% (12)	100% (7)	100% (8)	
Neuronal necrosis and loss	0	100% (3)	0%	0%	0%	*p* < 0.001
	1	0%	83.3% (10)	28.6% (2)	0%	
	2	0%	16.7% (2)	71.4% (5)	75% (6)	
	3	0%	0%	0%	25% (2)	
	Total	100% (3)	100% (12)	100% (7)	100% (8)	
Microglial lipofuscin deposits	0	100% (3)	0%	0%	0%	*p* < 0.001
	1	0%	83.3% (10)	14.3% (1)	0%	
	2	0%	16.7% (2)	71.4% (5)	25% (2)	
	3	0%	0	14.3% (1)	75% (6)	
	Total	100% (3)	100% (12)	100% (7)	100% (8)	
Perivascular microglia	0	100% (3)	0%	0%	0%	*p* < 0.001
	1	0%	100% (12)	71.4% (5)	37.5% (3)	
	2	0%	0%	28.6% (2)	37.5% (3)	
	3	0%	0%	0%	25% (2)	
	Total	100% (3)	100% (12)	100% (7)	100% (8)	
Spheroids	0	100% (3)	75% (9)	85.7% (6)	62.5% (5)	Non-significant
	1	0%	16.7% (2)	14.3% (1)	12.5% (1)	
	2	0%	8.3% (1)	0%	25% (2)	
	3	0%	0%	0%	0%	
	Total	100% (3)	100% (12)	100% (7)	100% (8)	
Lafora-like bodies	0	100% (3)	8.3% (1)	0%	0%	*p* < 0.001
	1	0%	66.7% (8)	57.1% (4)	12.5% (1)	
	2	0%	25% (3)	28.6% (2)	62.5% (5)	
	3	0%	0%	14.3% (1)	25% (2)	
	Total	100% (3)	100%	100%	100%	
H-E-positive bodies	0	100% (3)	58.4% (7)	85.7% (6)	0%	*p* = 0.014
	1	0%	33.3% (4)	0%	62.5% (5)	
	2	0%	8.3% (1)	14.3% (1)	37.5% (3)	
	3	0%	0%	0%	0%	
	Total	100% (3)	100% (12)	100% (7)	100% (8)	
Neuroaxonal degeneration	0	100% (3)	0%	0%	0%	*p* < 0.001
	1	0%	83.3% (10)	57.1% (4)	0%	
	2	0%	16.7% (2)	42.9% (3)	25% (2)	
	3	0%	0%	0%	75% (6)	
	Total	100% (3)	100% (12)	100% (7)	100% (8)	
GM and WM vacuolation	0	100% (3)	0%	0%	0%	*p* < 0.001
	1	0%	66.7% (8)	57.1% (4)	12.5% (1)	
	2	0%	25% (3)	42.9% (3)	50% (4)	
	3	0%	8.3% (1)	0%	37.5% (3)	
	Total	100% (3)	100% (12)	100% (7)	100% (8)	
GM and WM vascular hyalinosis	0	100% (3)	83.3% (10)	42.9% (3)	25% (2)	*p* = 0.040
	1	0%	16.7% (2)	57.1% (4)	50% (4)	
	2	0%	0%	0%	25% (2)	
	3	0%	0%	0%	0%	
	Total	100% (3)	100% (12)	100% (7)	100% (8)	
GM and WM vascular fibrosis	0	100% (3)	66.7% (8)	28.6% (2)	25% (2)	*p* = 0.055
	1	0%	33.3% (4)	71.4% (5)	75% (6)	
	2	0%	0%	0%	0%	
	3	0%	0%	0%	0%	
	Total	100% (3)	100% (12)	100% (7)	100% (8)	
Leptomeningeal fibrosis	0	100% (3)	33.3% (4)	0%	0%	*p* < 0.001
	1	0%	58.4% (7)	42.9% (3)	12.5% (1)	
	2	0%	8.3% (1)	57.1% (4)	37.5% (3)	
	3	0%	0%	0%	4	
	Total	100% (3)	100% (12)	100% (7)	100% (8)	
LV fibrosis	0	100% (3)	33.3% (4)	0%	0%	*p* < 0.001
	1	0%	41.7% (5)	14.3% (1)	0%	
	2	0%	25% (3)	85.7% (6)	62.5% (5)	
	3	0%	0%	0%	37.5% (3)	
	Total	100% (3)	100% (12)	100% (7)	100% (8)	
LV hyalinosis	0	100% (3)	100% (12)	85.7% (6)	62.5% (5)	Non-significant
	1	0%	0%	14.3% (1)	12.5% (1)	
	2	0%	0%	0%	25% (2)	
	3	0%	0%	0%	0%	
	Total	100% (3)	100% (12)	100% (7)	100% (8)	
LV calcification	0	100% (3)	75% (9)	71.4% (5)	75% (6)	Non-significant
	1	0%	25% (3)	28.6% (2)	25% (2)	
	2	0%	0%	0%	0%	
	3	0%	0%	0%	0%	
	Total	100% (3)	100% (12)	100% (7)	100% (8)	
CPV and CP epithelial fibrosis	0	100% (3)	25% (3)	0%	0%	*p* < 0.001
	1	0%	66.7% (8)	28.6% (2)	0%	
	2	0%	8.3% (1)	42.8% (3)	37.5% (3)	
	3	0%	0%	28.6% (2)	62.5% (5)	
	Total	100% (3)	100% (12)	100% (7)	100% (8)	
CPV hyalinosis	0	100% (3)	100% (12)	71.4% (5)	62.5% (5)	Non-significant
	1	0%	0%	28.6% (2)	37.5% (3)	
	2	0%	0%	0%	0%	
	3	0%	0%	0%	0%	
	Total	100% (3)	100% (12)	100% (7)	100% (8)	
Hemorrhages	0	100% (3)	75% (9)	100% (7)	62.5% (5)	Non-significant
	1	0%	25% (3)	0%	37.5% (3)	
	2	0%	0%	0%	0%	
	3	0%	0%	0%	0%	
	Total	100% (3)	100% (12)	100% (7)	100% (8)	

Kruskal–Wallis test. Neuronophagia: Controls vs. Mature Group: *p* = 0.044, Controls vs. Senior group: *p* = 0.021, Controls vs. Geriatric group: *p* < 0.001, Mature vs. Geriatric group: *p* < 0.001, Senior vs. Geriatric group: *p* = 0.008; Satellitosis: Controls vs. Mature Group: *p* = 0.043, Controls vs. Senior group: *p* = 0.016, Controls vs. Geriatric group: *p* < 0.001, Mature vs. Geriatric group: *p* = 0.003, Senior vs. Geriatric group: *p* = 0.049; Chromatolysis: Controls vs. Mature Group: *p* = 0.063 ^#^, Controls vs. Senior group: *p* = 0.005, Controls vs. Geriatric group: *p* < 0.001, Mature vs. Geriatric group: *p* = 0.001; Neuronal lipofuscin deposits: Controls vs. Mature Group: *p* = 0.080 ^#^, Controls vs. Senior group: *p* = 0.005, Controls vs. Geriatric group: *p* < 0.001, Mature vs. Senior group: *p* = 0.087 ^#^, Mature vs. Geriatric group: *p* < 0.001; Neuronal necrosis and loss: Controls vs. Mature Group: *p* = 0.068 ^#^, Controls vs. Senior group: *p* = 0.003, Controls vs. Geriatric group: *p* < 0.001, Mature vs. Senior group: *p* = 0.077 ^#^, Mature vs. Geriatric group: *p* < 0.001; Microglial lipofuscin deposits: Controls vs. Senior group: *p* = 0.004, Controls vs. Geriatric group: *p* < 0.001, Mature vs. Senior group: *p* = 0.044, Mature vs. Geriatric group: *p* < 0.001; Perivascular microglia: Controls vs. Mature Group: *p* = 0.015, Controls vs. Senior group: *p* = 0.003, Controls vs. Geriatric group: *p* < 0.001, Mature vs. Geriatric group: *p* = 0.010; Lafora-like bodies: Controls vs. Mature Group: *p* = 0.045, Controls vs. Senior group: *p* = 0.012, Controls vs. Geriatric group: *p* < 0.001, Mature vs. Geriatric group: *p* = 0.012; H-E-positive bodies: Controls vs. Geriatric group: *p* = 0.005, Mature vs. Geriatric group: *p* = 0.010, Senior vs. Geriatric group: *p* = 0.003; Neuroaxonal degeneration: Controls vs. Mature Group: *p* = 0.054^3^, Controls vs. Senior group: *p* = 0.022, Controls vs. Geriatric group: *p* < 0.001, Mature vs. Geriatric group: *p* < 0.001, Senior vs. Geriatric group: *p* = 0.011; GM and WM vacuolation: Controls vs. Mature Group: *p* = 0.020, Controls vs. Senior group: *p* = 0.023, Controls vs. Geriatric group: *p* < 0.001, Mature vs. Geriatric group: *p* = 0.028, Senior vs. Geriatric group: *p* = 0.069 ^#^; GM and WM vascular hyalinosis: Controls vs. Geriatric group: *p* = 0.020, Mature vs. Geriatric group: *p* = 0.005; Leptomeningeal fibrosis: Controls vs. Senior group: *p* = 0.013, Controls vs. Geriatric group: *p* < 0.001, Mature vs. Senior group: *p* = 0.057 *^#^*, Mature vs. Geriatric group: *p* < 0.001; LV fibrosis: Controls vs. Senior group: *p* = 0.008, Controls vs. Geriatric group: *p* < 0.001, Mature vs. Senior group: *p* = 0.036, Mature vs. Geriatric group: *p* < 0.001; CPV and CP epithelial fibrosis: Controls vs. Senior group: *p* = 0.006, Controls vs. Geriatric group: *p* < 0.001, Mature vs. Senior group: *p* = 0.020, Mature vs. Geriatric group: *p* < 0.001.

**Table 5 brainsci-13-01115-t005:** Iron deposition scoring in the brains of aged cats.

Groups						
Investigated Parameter	Grading Score	Controls	Mature	Senior	Geriatric	Significance (Chi Square)
Temporal GM and WM cells	0	0%	33.3% (4)	0%	12.5% (1)	Non-significant
	1	66.7% (2)	41.7% (5)	85.7% (6)	25% (2)	
	2	33.3% (1)	16.7% (2)	0%	75% (6)	
	3	0%	8.3% (1)	14.3% (1)	25% (2)	
	Total	100% (3)	100% (12)	100% (7)	100% (8)	
Temporal GM and WM IPs	0	0%	0%	0%	0%	Non-significant
	1	66.7% (2)	50% (6)	42.8% (3)	12.5% (1)	
	2	33.3% (1)	41.7% (5)	28.6% (2)	37.5% (3)	
	3	0%	8.3% (1)	28.6% (2)	50% (4)	
	Total	100% (3)	100% (12)	100% (7)	100% (8)	
Temporal WM MF *	0	100% (3)	50% (6)	42.8% (3)	25% (2)	Non-significant
	1	0%	41.7% (5)	14.3% (1)	37.5% (3)	
	2	0%	0%	14.3% (1)	12.5% (1)	
	3	0%	8.3% (1)	28.6% (2)	25% (2)	
	Total	100% (3)	100% (12)	100% (7)	100% (8)	
Hippocampal cells	0	0%	8.3% (1)	0%	0%	Non-significant
	1	66.7% (2)	66.7% (8)	71.4% (5)	37.5% (3)	
	2	0%	0%	14.3% (1)	37.5% (3)	
	3	33.3% (1)	25% (3)	14.3% (1)	25% (2)	
	Total	100% (3)	100% (12)	100% (7)	100% (8)	
Hippocampal IPs	0	66.7% (2)	16.7% (2)	14.3% (1)	0%	Non-significant
	1	33.3% (1)	75% (9)	57.1% (4)	75% (6)	
	2	0%	8.3% (1)	28.6% (2)	12.5% (1)	
	3	0%	0%	0%	12.5% (1)	
	Total	100% (3)	100% (12)	100% (7)	100% (8)	
Thalamic cells	0	0%	33.3% (4)	14.3% (1)	0%	Non-significant
	1	100% (3)	66.7% (8)	85.7% (6)	75% (6)	
	2	0%	0%	0%	12.5% (1)	
	3	0%	0%	0%	12.5% (1)	
	Total	100% (3)	100% (12)	100% (7)	100% (8)	
Thalamic IPs	0	100% (3)	0%	0%	0%	*p* = 0.025
	1	0%	66.7% (8)	57.1% (4)	12.5% (1)	
	2	0%	33.3% (4)	14.3% (1)	25% (2)	
	3	0%	0%	28.6% (2)	62.5% (5)	
	Total	100% (3)	100% (12)	100% (7)	100% (8)	
Striatal cells	0	100% (3)	50% (6)	42.9% (3)	37.5% (3)	Non-significant
	1	0%	41.7% (5)	57.1% (4)	50% (4)	
	2	0%	8.3% (1)	0%	12.5% (1)	
	3	0%	0%	0%	0%	
	Total	100% (3)	100% (12)	100% (7)	100% (8)	
Striatal IPs	0	33.3% (1)	16.7% (2)	28.6% (2)	12.5% (1)	Non-significant
	1	66.7%(2)	83.3% (10)	71.4% (5)	75% (6)	
	2	0%	0%	0%	12.5% (1)	
	3	0%	0%	0%	0%	
	Total	100% (3)	100% (12)	100% (7)	100% (8)	
Frontal GM and WM cells	0	33.3% (1)	16.7% (2)	14.3% (1)	0%	Non-significant
	1	66.7%(2)	66.7% (8)	71.4% (5)	62.5% (5)	
	2	0%	8.3% (1)	14.3% (1)	37.5% (3)	
	3	0%	8.3% (1)	0%	0%	
	Total	100% (3)	100% (12)	100% (7)	100% (8)	
Frontal GM and WM IPs	0	0%	16.7% (2)	0%	0%	*p* = 0.088 ^#^
	1	100% (3)	58.4% (7)	57.1% (4)	12.5% (1)	
	2	0%	16.7% (2)	42.9% (3)	50% (4)	
	3	0%	8.3% (1)	0%	37.5% (3)	
	Total	100% (3)	100% (12)	100% (7)	100% (8)	
Frontal WM MF *	0	100% (3)	41.7% (5)	42.9% (3)	12.5% (1)	Non-significant
	1	0%	41.7% (5)	42.9% (3)	25% (2)	
	2	0%	16.6% (2)	14.2% (1)	37.5% (3)	
	3	0%	0%	0%	25% (2)	
	Total	100% (3)	100% (12)	100% (7)	100% (8)	
Cerebellar GM, WM, and DN * cells	0	0%	0%	14.3% (1)	0%	Non-significant
	1	100% (3)	66.7% (8)	28.6% (2)	37.5% (3)	
	2	0%	25% (3)	57.1% (4)	37.5% (3)	
	3	0%	8.3% (1)	0%	25% (2)	
	Total	100% (3)	100% (12)	100% (7)	100% (8)	
Cerebellar GM, WM, and DN * IPs	0	33.3% (1)	0%	0%	0%	Non-significant
	1	33.3% (1)	58.3% (7)	14.2% (1)	37.5% (3)	
	2	33.4% (1)	25% (3)	42.9% (3)	25% (2)	
	3	0%	16.7% (2)	42.9% (3)	37.5% (3)	
	Total	100% (3)	100% (12)	100% (7)	100% (8)	
Cerebellar WM MF *	0	66.7% (2)	25% (3)	14.3% (1)	0%	*p* = 0.002
	1	33.3% (1)	50% (6)	57.1% (4)	0%	
	2	0%	25% (3)	14.3% (1)	12.5% (1)	
	3	0%	0%	14.3% (1)	87.5% (7)	
	Total	100% (3)	100% (12)	100% (7)	100% (8)	

Kruskal–Wallis test. Thalamic IPs: Controls vs. Geriatric group: *p* = 0.008, Mature vs. Geriatric group: *p* = 0.004, Senior vs. Geriatric group: *p* = 0.069 ^#^; Frontal GM and WM IPs: Controls vs. Geriatric group: *p* = 0.018, Mature vs. Geriatric group: *p* = 0.003, Senior vs. Geriatric group: *p* = 0.063; Frontal WM MF: Controls vs. Geriatric group: *p* = 0.005, Mature vs. Geriatric group: *p* = 0.038, Senior vs. Geriatric group: *p* = 0.057; Cerebellar WM MF: Controls vs. Geriatric group: *p* < 0.001, Mature vs. Geriatric group: *p* < 0.001, Senior vs. Geriatric group: *p* = 0.007. * MF: myelinated fibers; DN: dentate nucleus.

**Table 6 brainsci-13-01115-t006:** Scoring of MT-I/II immunolabeling in cats of different ages.

Groups						
Investigated Parameter	Grading Score	Controls	Mature	Senior	Geriatric	Significance (Chi Square)
Temporal GM	0	33.3% (1)	0%	0%	0%	*p* < 0.001
	1	66.7% (2)	100% (12)	71.4% (5)	12.5% (1)	
	2	0%	0%	28.6% (2)	50% (4)	
	3	0%	0%	0%	37.5% (3)	
	Total	100% (3)	100% (12)	100% (7)	100% (8)	
Temporal WM	0	33.3% (1)	0%	0%	0%	*p* < 0.001
	1	66.7% (2)	91.7% (11)	14.3% (1)	0%	
	2	0%	8.3% (1)	71.4% (5)	25% (2)	
	3	0%	0%	14.3% (1)	75% (6)	
	Total	100% (3)	100% (12)	100% (7)	100% (8)	
Temporal LV	0	0%	16.7% (2)	0%	0%	Non-significant
	1	100% (3)	83.3% (10)	100% (7)	100% (8)	
	2	0%	0%	0%	0%	
	3	0%	0%	0%	0%	
	Total	100% (3)	100% (12)	100% (7)	100% (8)	
Temporal GM and WM vessels	0	0%	25% (3)	0%	12.5% (1)	Non-significant
	1	100% (3)	75% (6) (9)	100% (7)	87.5% (7)	
	2	0%	0%	0%	0%	
	3	0%	0%	0%	0%	
	Total	100% (3)	100% (12)	100% (7)	100% (8)	
Hippocampal GM	0	33.3% (1)	0%	0%	0%	*p* = 0.001
	1	33.3% (1)	16.7% (2)	14.3% (1)	0%	
	2	33.3% (1)	83.3% (10)	71.4% (5)	25% (2)	
	3	0%	0%	14.3% (1)	75% (6)	
	Total	100% (3)	100% (12)	100% (7)	100% (8)	
Hippocampal WM	0	33.3% (1)	0%	0%	0%	*p* = 0.041
	1	33.3% (1)	16.7% (2)	14.3% (1)	0%	
	2	33.3% (1)	83.3% (10)	71.4% (5)	62.5% (5)	
	3	0%	0%	14.3% (1)	37.5% (3)	
	Total	100% (3)	100% (12)	100% (7)	100% (8)	
Hippocampal vessels	0	100% (3)	83.3% (10)	57.1% (4)	25% (2)	*p* = 0.030
	1	0%	16.7% (2)	42.9% (3)	75% (6)	
	2	0%	0%	0%	0%	
	3	0%	0%	0%	0%	
	Total	100% (3)	100% (12)	100% (7)	100% (8)	
Thalamus	0	33.3% (1)	0%	0%	0%	*p* < 0.001
	1	33.3% (1)	75% (9)	42.9% (3)	0%	
	2	33.3% (1)	25% (3)	57.1% (4)	25% (2)	
	3	0%	0%	0%	75% (6)	
	Total	100% (3)	100% (12)	100% (7)	100% (8)	
Thalamic vessels	0	33.3% (1)	83.3% (10)	28.6% (2)	25% (2)	*p* = 0.030
	1	66.7% (2)	16.7% (2)	71.4% (5)	75% (6)	
	2	0%	0%	0%	0%	
	3	0%	0%	0%	0%	
	Total	100% (3)	100% (12)	100% (7)	100% (8)	
Striatum	0	66.7% (2)	8,3%	0%	0%	*p* < 0.001
	1	33.3% (1)	91.7%	85.7% (6)	25% (2)	
	2	0%	0%	14.3% (1)	75% (6)	
	3	0%	0%	0%	0%	
	Total	100% (3)	100% (12)	100% (7)	100% (8)	
Striatal vessels	0	66.7% (2)	83.3% (10)	28.6% (2)	25% (2)	*p* = 0.031
	1	33.3% (1)	16.7% (2)	71.4% (5)	75% (6)	
	2	0%	0%	0%	0%	
	3	0%	0%	0%	0%	
	Total	100% (3)	100% (12)	100% (7)	100% (8)	
Internal capsule	0	0%	0%	0%	0%	*p* = 0.017
	1	66.7% (2)	66.7% (8)	28.6% (2)	0%	
	2	33.3% (1)	33.3% (4)	71.4% (5)	100% (8)	
	3	0%	0%	0%	0%	
	Total	100% (3)	100% (12)	100% (7)	100% (8)	
Frontal GM	0	0%	0%	0%	0%	*p* = 0.060 ^#^
	1	66.7% (2)	58.3% (7)	42.8% (3)	0%	
	2	33.3% (1)	41.7% (5)	28.6% (2)	50% (4)	
	3	0%	0%	28.6% (2)	50% (4)	
	Total	100% (3)	100% (12)	100% (7)	100% (8)	
Frontal WM	0	0%	0%	0%	0%	*p* = 0.005
	1	66.7% (2)	50% (6)	0%	0%	
	2	33.3% (1)	41.7% (5)	71.4% (5)	25% (2)	
	3	0%	8,3% (1)	28.6% (2)	75% (6)	
	Total	100% (3)	100% (12)	100% (7)	100% (8)	
Frontal LC	0	100% (3)	83.3% (10)	71.4% (5)	12.5% (1)	*p* = 0.005
	1	0%	16.7% (2)	28.6% (2)	87.5% (7)	
	2	0%	0%	0%	0%	
	3	0%	0%	0%	0%	
	Total	100% (3)	100% (12)	100% (7)	100% (8)	
Frontal LV	0	0%	8,3% (1)	0%	0%	Non-significant
	1	100% (3)	91.7% (11)	100% (7)	100% (8)	
	2	0%	0%	0%	0%	
	3	0%	0%	0%	0%	
	Total	100% (3)	100% (12)	100% (7)	100% (8)	
Frontal GM and WM vessels	0	66.7% (2)	83.3% (10)	57.1% (4)	50% (4)	Non-significant
	1	33.3% (1)	16.7% (2)	42.9% (3)	50% (4)	
	2	0%	0%	0%	0%	
	3	0%	0%	0%	0%	
	Total	100% (3)	100% (12)	100% (7)	100% (8)	
CP	0	66.7% (2)	66.7% (8)	0%	0%	*p* = 0.002
	1	33.3% (1)	33.3% (4)	100% (7)	100% (8)	
	2	0%	0%	0%	0%	
	3	0%	0%	0%	0%	
	Total	100% (3)	100% (12)	100% (7)	100% (8)	
CPV	0	0%	41.7% (5)	0%	0%	*p* = 0.029
	1	100% (3)	58.3% (7)	100% (7)	100% (8)	
	2	0%	0%	0%	0%	
	3	0%	0%	0%	0%	
	Total	100% (3)	100% (12)	100% (7)	100% (8)	
Ependyma	0	33.3% (1)	16.7% (2)	0%	0%	Non-significant
	1	66.7% (2)	83.3% (10)	100% (7)	100% (8)	
	2	0%	0%	0%	0%	
	3	0%	0%	0%	0%	
	Total	100% (3)	100% (12)	100% (7)	100% (8)	

Kruskal–Wallis test. Temporal GM: Controls vs. Geriatric group: *p* < 0.001, Mature vs. Geriatric group: *p* < 0.001, Senior vs. Geriatric group *p* = 0.014; Temporal WM: Controls vs. Geriatric group: *p* < 0.001, Mature vs. Geriatric group: *p* < 0.001, Controls vs. Senior group: *p* = 0.022, Mature vs. Senior group *p* = 0.014; Hippocampal GM: Controls vs. Geriatric group: *p* < 0.001, Mature vs. Geriatric group: *p* = 0.001, Senior vs. Geriatric group: *p* = 0.022; Hippocampal WM: Controls vs. Geriatric group: *p* = 0.004, group 2 vs. group 4 *p* = 0.037, Controls vs. Senior group: *p* = 0.062 ^#^; Thalamus: Controls vs. Geriatric group: *p* = 0.004, Mature vs. Geriatric group: *p* < 0.001, Senior vs. Geriatric group: *p* = 0.010; STR: Controls vs. Geriatric group: *p* < 0.001, Mature vs. Geriatric group: *p* < 0.001, Controls vs. Senior group: *p* = 0.057, Senior vs. Geriatric group: *p* = 0.030; IC: Controls vs. Geriatric group: *p* = 0.048, Mature vs. Geriatric group: *p* = 0.003.

**Table 7 brainsci-13-01115-t007:** Grading of GFAP immunolabeling.

Groups						
Investigated Parameter	Grading Score	Controls	Mature	Senior	Geriatric	Significance (Chi Square)
Temporal GM	0	100% (3)	0%	0%	0%	*p* < 0.001
	1	0%	16.7% (2)	0%	0%	
	2	0%	75% (9)	42.9% (3)	12.5% (1)	
	3	0%	8.3% (1)	57.1% (4)	87.5% (7)	
	Total	100% (3)	100% (12)	100% (7)	100% (8)	
Temporal WM	0	100% (3)	0%	0%	0%	*p* < 0.001
	1	0%	8.3% (1)	0%	0%	
	2	0%	91.7% (11)	28.6% (2)	0%	
	3	0%	0%	71.4% (5)	100%	
	Total	100% (3)	100% (12)	100% (7)	100% (8)	
Temporal GL	0	100% (3)	0%	0%	0%	*p* < 0.001
	1	0%	58.3% (7)	0%	0%	
	2	0%	41.7% (5)	85.7% (6)	50% (4)	
	3	0%	0%	14.3% (1)	50% (4)	
	Total	100% (3)	100% (12)	100% (7)	100% (8)	
Hippocampal GM	0	100% (3)	0%	0%	0%	*p* < 0.001
	1	0%	16.7% (2)	14.2% (1)	0%	
	2	0%	75% (9)	42.9% (3)	12.5% (1)	
	3	0%	8.3% (1)	42.9% (3)	87.5% (7)	
	Total	100% (3)	100% (12)	100% (7)	100% (8)	
Hippocampal WM	0	100% (3)	0%	0%	0%	*p* < 0.001
	1	0%	16.7% (2)	0%	0%	
	2	0%	83.3% (10)	85.7% (6)	37.5% (3)	
	3	0%	0%	14.3% (1)	62.5% (5)	
	Total	100% (3)	100% (12)	100% (7)	100% (8)	
Hippocampal GL	0	100% (3)	100%	0%	100%	*p* < 0.001
	1	0%	66.7% (8)	0%	0%	
	2	0%	33.3% (4)	100% (7)	62.5% (5)	
	3	0%	0%	0%	37.5% (3)	
	Total	100% (3)	100% (12)	100% (7)	100% (8)	
Frontal GM	0	100% (3)	0%	0%	0%	*p* < 0.001
	1	0%	58.3% (7)	0%	0%	
	2	0%	41.7% (5)	85.7% (6)	37.5% (3)	
	3	0%	0%	14.3% (1)	62.5% (5)	
	Total	100% (3)	100% (12)	100% (7)	100% (8)	
Frontal WM	0	100% (3)	0%	0%	0%	*p* < 0.001
	1	0%	16.7% (2)	0%	0%	
	2	0%	75% (9)	71.4% (5)	25% (2)	
	3	0%	8.3% (1)	28.6% (2)	75% (6)	
	Total	100% (3)	100% (12)	100% (7)	100% (8)	
Frontal GL	0	100% (3)	0%	0%	0%	*p* < 0.001
	1	0%	50% (6)	28.6% (2)	0%	
	2	0%	50% (6)	71.4% (5)	50% (4)	
	3	0%	0%	0%	50% (4)	
	Total	100% (3)	100% (12)	100% (7)	100% (8)	
Cerebellar GM	0	100% (3)	0%	0%	0%	*p* < 0.001
	1	0%	16.7% (2)	0%	0%	
	2	0%	75% (9)	57.1% (4)	12.5% (1)	
	3	0%	8.3% (1)	42.9% (3)	87.5% (7)	
	Total	100% (3)	100% (12)	100% (7)	100% (8)	
Cerebellar WM	0	100% (3)	0%	0%	0%	*p* < 0.001
	1	0%	8.3% (1)	0%	0%	
	2	0%	91.7% (11)	85.7% (6)	12.5% (1)	
	3	0%	0%	14.3% (1)	87.5% (7)	
	Total	100% (3)	100% (12)	100% (7)	100% (8)	
Cerebellar GL	0	100% (3)	0%	0%	0%	*p* < 0.001
	1	0%	66.7% (8)	0%	0%	
	2	0%	33.3% (4)	85.7% (6)	37.5% (3)	
	3	0%	0%	14.3% (1)	62.5% (5)	
	Total	100% (3)	100% (12)	100% (7)	100% (8)	

Kruskal–Wallis test. Temporal GM: Controls vs. Mature Group: *p* = 0.062 ^#^, Controls vs. Senior group: *p* = 0.002, Controls vs. Geriatric group: *p* < 0.001, Mature vs. Senior group: *p* = 0.057 ^#^, Mature vs. Geriatric group: *p* = 0.003; Temporal WM: Controls vs. Senior group: *p* = 0.001, Controls vs. Geriatric group: *p* < 0.001, Mature vs. Senior group: *p* = 0.010, Mature vs. Geriatric group: *p* < 0.001; Temporal GL: Controls vs. Mature Group: *p* = 0.069 ^#^, Controls vs. Senior group: *p* = 0.002, Controls vs. Geriatric group: *p* < 0.001, Mature vs. Senior group: *p* = 0.043, Mature vs. Geriatric group: *p* = 0.002; Hippocampal GM: Controls vs. Mature Group: *p* = 0.044, Controls vs. Senior group: *p* = 0.008, Controls vs. Geriatric group: *p* < 0.001, Mature vs. Geriatric group: *p* = 0.004; Hippocampal WM: Controls vs. Mature Group: *p* = 0.021, Controls vs. Senior group: *p* = 0.005, Controls vs. Geriatric group: *p* < 0.001, Mature vs. Geriatric group: *p* = 0.006; Hippocampal GL: Controls vs. Mature Group: *p* = 0.066 ^#^, Controls vs. Senior group: *p* = 0.002, Controls vs. Geriatric group: *p* < 0.001, Mature vs. Senior group: *p* = 0.036, Mature vs. Geriatric group: *p* = 0.002; Frontal GM: Controls vs. Mature Group: *p* = 0.078 ^#^, Controls vs. Senior group: *p* = 0.003, Controls vs. Geriatric group: *p* < 0.001, Mature vs. Senior group: *p* = 0.054 ^#^, Mature vs. Geriatric group: *p* = 0.001; Frontal WM: Controls vs. Mature Group: *p* = 0.032, Controls vs. Senior group: *p* = 0.006, Controls vs. Geriatric group: *p* < 0.001, Mature vs. Geriatric group: *p* = 0.007; Frontal GL: Controls vs. Mature Group: *p* = 0.032, Controls vs. Senior group: *p* = 0.015, Controls vs. Geriatric group: *p* < 0.001, Mature vs. Geriatric group: *p* = 0.005, Senior vs. Geriatric group: *p* = 0.056 ^#^; Cerebellar GM: Controls vs. Mature Group: *p* = 0.051 ^#^, Controls vs. Senior group, *p* = 0.005; Controls vs. Geriatric group: *p* < 0.001, Mature vs. Geriatric group: *p* = 0.002; Cerebellar WM: Controls vs. Mature Group: *p* = 0.031, Controls vs. Senior group: *p* = 0.012, Controls vs. Geriatric group: *p* < 0.001, Mature vs. Geriatric group: *p* < 0.001, Senior vs. Geriatric group: *p* = 0.017; Cerebellar GL: Controls vs. Mature Group: *p* = 0.093 ^#^, Controls vs. Senior group: *p* = 0.002, Controls vs. Geriatric group: *p* < 0.001, Mature vs. Senior group: *p* = 0.035, Mature vs. Geriatric group: *p* < 0.001.

**Table 8 brainsci-13-01115-t008:** Scoring of Aβ immunolabeling in the brain of cats of different ages.

Groups						
Investigated Parameter	Grading Score	Controls	Mature	Senior	Geriatric	Significance (Chi Square)
Temporal leptomeningeal CAA	0	100% (3)	0%	0%	0%	*p* < 0.001
	1	0%	50% (6)	14.3% (1)	62.5% (5)	
	2	0%	41.7% (5)	71.4%	25% (2)	
	3	0%	8.3% (1)	14.3% (1)	12.5% (1)	
	Total	100% (3)	100% (12)	100% (7)	100% (8)	
Temporal GM and WM CAA	0	100% (3)	0%	0%	0%	*p* < 0.001
	1	0%	41.7% (5)	71.4%	87.5% (7)	
	2	0%	25% (3)	14.3% (1)	0%	
	3	0%	33.3% (4)	14.3% (1)	12.5% (1)	
	Total	100% (3)	100% (12)	100% (7)	100% (8)	
Temporal GM neurons	0	100% (3)	33.3% (4)	42.9% (3)	75% (6)	*p* = 0.095 ^#^
	1	0%	66.7% (8)	57.1% (4)	25% (2)	
	2	0%	0%	0%	0%	
	3	0%	0%	0%	0%	
	Total	100% (3)	100% (12)	100% (7)	100% (8)	
Temporal GM and WM SPs	0	100% (3)	50% (6)	0%	0%	*p* < 0.001
	1	0%	25% (3)	14.2% (1)	0%	
	2	0%	0%	42.9% (3)	0%	
	3	0%	25% (3)	42.9% (3)	100% (8)	
	Total	100% (3)	100% (12)	100% (7)	100% (8)	
Hippocampal neurons	0	100% (3)	8.3% (1)	14.3% (1)	25% (2)	*p* = 0.002
	1	0%	83.4% (10)	28.6% (2)	25% (2)	
	2	0%	8.3% (1)	57.1% (4)	50% (4)	
	3	0%	0%	0%	0%	
	Total	100% (3)	100% (12)	100% (7)	100% (8)	
Hippocampal SPs	0	100% (3)	83.3% (10)	71.4%	25% (2)	Non-significant
	1	0%	16.7% (2)	28.6% (2)	37.5% (3)	
	2	0%	0%	0%	12.5% (1)	
	3	0%	0%	0%	25% (2)	
	Total	100% (3)	100% (12)	100% (7)	100% (8)	
Thalamic neurons	0	100% (3)	16.7% (2)	14.3% (1)	37.5% (3)	0.029
	1	0%	8.3% (1)	85.7% (6)	62.5% (5)	
	2	0%	0%	0%	0%	
	3	0%	0%	0%	0%	
	Total	100% (3)	100% (12)	100% (7)	100% (8)	
Thalamic SPs	0	100% (3)	100% (12)	71.4% (5)	37.5% (3)	*p* = 0.092 ^#^
	1	0%	0%	0%	12.5% (1)	
	2	0%	0%	0%	25% (2)	
	3	0%	0%	28.6% (2)	25% (2)	
	Total	100% (3)	100% (12)	100% (7)	100% (8)	
Striatal neurons	0	100% (3)	25% (3)	28.6% (2)	37.5% (3)	Non-significant
	1	0%	75% (9)	57.1% (4)	62.5% (5)	
	2	0%	0%	14.3% (1)	0%	
	3	0%	0%	0%	0%	
	Total	100% (3)	100% (12)	100% (7)	100% (8)	
Striatal SPs	0	100% (3)	100% (12)	57.1% (4)	25% (2)	*p* = 0.047
	1	0%	0%	28.6% (2)	37.5% (3)	
	2	0%	0%	0%	25% (2)	
	3	0%	0%	14.3% (1)	12.5% (1)	
	Total	100% (3)	100% (12)	100% (7)	100% (8)	
CPV amyloid deposits	0	100% (3)	0%	0%	0%	*p* < 0.001
	1	0%	50% (6)	14.3% (1)	25% (2)	
	2	0%	33.3% (4)	71.4% (5)	37.5% (3)	
	3	0%	16.7% (2)	14.3% (1)	37.5% (3)	
	Total	100% (3)	100% (12)	100% (7)	100% (8)	
CP epithelial cells	0	100% (3)	0%	0%	0%	*p* < 0.001
	1	0%	33.3% (4)	0%	0%	
	2	0%	41.7% (5)	57.1% (4)	37.5% (3)	
	3	0%	25% (3)	42.9% (3)	62.5% (5)	
	Total	100% (3)	100% (12)	100% (7)	100% (8)	
Frontal leptomeningeal CAA	0	100% (3)	0%	0%	0%	*p* < 0.001
	1	0%	8.3% (1)	0%	0%	
	2	0%	41.7%	28.6% (2)	25% (2)	
	3	0%	50% (6)%	71.4%	75% (6)	
	Total	100% (3)	100% (12)	100% (7)	100% (8)	
Frontal GM and WM CAA	0	100% (3)	8.3% (1)	0%	0%	*p* = 0.002
	1	0%	58.3% (7)	28.6% (2)	25% (2)	
	2	0%	16.7% (2)	42.8% (3)	50% (4)	
	3	0%	16.7% (2)	28.6% (2)	25% (2)	
	Total	100% (3)	100% (12)	100% (7)	100% (8)	
Frontal GM neurons	0	100% (3)	8.3% (1)	0%	25% (2)	*p* = 0.021
	1	0%	75% (9)	85.7% (6)	62.5% (5)	
	2	0%	16.7% (2)	14.3% (1)	12.5% (1)	
	3	0%	0%	0%	0%	
	Total	100% (3)	100% (12)	100% (7)	100% (8)	
Frontal GM and WM SPs	0	100% (3)	58.3% (7)	28.6% (2)	0%	*p <* 0.001
	1	0%	41.7% (5)	28.6% (2)	0%	
	2	0%	0%	14.3% (1)	0%	
	3	0%	0%	28.6% (2)	100% (8)	
	Total	100% (3)	100% (12)	100% (7)	100% (8)	
Cerebellar leptomeningeal CAA	0	100% (3)	8.3% (1)	0%	0%	*p* = 0.002
	1	0%	41.7% (5)	42.8% (3)	12.5% (1)	
	2	0%	16.7% (2)	28.6% (2)	50% (4)	
	3	0%	33.3% (4)	28.6% (2)	37.5% (3)	
	Total	100% (3)	100%	100%	100%	
Cerebellar GM and WM CAA	0	100% (3)	8.3% (1)	0%	0%	*p* < 0.001
	1	0%	50% (6)	28.6% (2)	12.5% (1)	
	2	0%	16.7% (2)	57.1% (4)	62.5% (5)	
	3	0%	25% (3)	14.3% (1)	25% (2)	
	Total	100% (3)	100% (12)	100% (7)	100% (8)	
Cerebellar GM and DN neurons	0	100% (3)	8.3% (1)	0%	0%	*p <* 0.001
	1	0%	91.7% (11)	100% (7)	75% (6)	
	2	0%	0%	0%	25% (2)	
	3	0%	0%	0%	0%	
	Total	100% (3)	100% (12)	100% (7)	100% (8)	

Kruskal–Wallis test. Temporal leptomeningeal CAA: Controls vs. Mature Group: *p* = 0.008, Controls vs. Senior group: *p* < 0.001, Controls vs. Geriatric group: *p* = 0.020; Temporal GM and WM CAA: Controls vs. Mature Group: *p* < 0.001, Controls vs. Senior group: *p* = 0.012, Controls vs. Geriatric group: *p* = 0.026; Temporal GM and WM SPs: Controls vs. Senior group: *p* = 0.020, Controls vs. Geriatric group: *p* < 0.001, Mature vs. Senior group: *p* = 0.077 ^#^, Mature vs. Geriatric group: *p* = 0.001; Hippocampal neurons: Controls vs. Mature Group: *p* = 0.045, Controls vs. Senior group: *p* = 0.006, Controls vs. Geriatric group: *p* = 0.014, Hippocampal SPs: Controls vs. Geriatric group: *p* = 0.015, Mature vs. Geriatric group: *p* = 0.004, Senior vs. Geriatric group: *p* = 0.032; Thalamic neurons: Controls vs. Mature Group: *p* = 0.006, Controls vs. Senior group: *p* = 0.008, Controls vs. Geriatric group: *p* = 0.048; Thalamic SPs: Controls vs. Geriatric group: *p* = 0.041, Mature vs. Geriatric group: *p* = 0.002; Striatal SPs: Controls vs. Geriatric group: *p* = 0.019, Mature vs. Senior group: *p* = 0.060 ^#^, Mature vs. Geriatric group: *p* < 0.001; CPV amyloid deposits: Controls vs. Mature Group: *p* = 0.018, Controls vs. Senior group: *p* = 0.005, Controls vs. Geriatric group: *p* = 0.002; CP epithelial cells: Controls vs. Mature Group: *p* = 0.028, Controls vs. Senior group: *p* = 0.004, Controls vs. Geriatric group: *p* < 0.001, Mature vs. Geriatric group: *p* = 0.065 ^#^; Frontal leptomeningeal CAA: Controls vs. Mature Group: *p* = 0.010, Controls vs. Senior group: *p* = 0.003, Controls vs. Geriatric group: *p* = 0.002; Frontal GM and WM CAA: Controls vs. Mature Group: *p* = 0.035, Controls vs. Senior group: *p* = 0.004, Controls vs. Geriatric group: *p* = 0.003; Frontal GM neurons: Controls vs. Mature Group: *p* = 0.003, Controls vs. Senior group: *p* = 0.004, Controls vs. Geriatric group: *p* = 0.023; Frontal GM and WM SPs: Controls vs. Mature Group: *p* = 0.046, Controls vs. Senior group: *p* = 0.078 ^#^, Controls vs. Geriatric group: *p* < 0.001, Mature vs. Geriatric group: *p* < 0.001, Senior vs. Geriatric group: *p* = 0.037; Cerebellar leptomeningeal CAA: Controls vs. Mature Group: *p* = 0.017, Controls vs. Senior group: *p* = 0.018, Controls vs. Geriatric group: *p* = 0.003; Cerebellar GM and WM CAA: Controls vs. Mature Group: *p* = 0.024, Controls vs. Senior group: *p* = 0.011, Controls vs. Geriatric group: *p* = 0.002; Cerebellar GM and DN neurons: Controls vs. Mature Group: *p* = 0.001, Controls vs. Senior group: *p* < 0.001, Controls vs. Geriatric group: *p* < 0.001.

**Table 9 brainsci-13-01115-t009:** Iron and zinc concentrations in brain tissues of all cats.

Groups	Case No	Fe (μg/g)	Zn (μg/g)
Controls	1	175.2	29.9
	2	225.3	18.5
	3	310.8	16.1
Mature	4	264.6	36.7
	5	252.6	17.9
	6	270.4	26.8
	7	269.6	28.8
	8	264.6	17.8
	9	313	18.7
	10	267	22.5
	11	333.8	35.8
	12	488.8	45.9
	13	248.3	19.4
	14	276.4	24.8
	15	243	31.59
Senior	16	227.8	32.38
	17	208.2	18.2
	18	268.7	24.8
	19	238.7	29.4
	20	318	22.2
	21	934.6	1202
	22	326	25.8
Geriatric	23	301.7	19.9
	24	260.7	18.4
	25	253.6	17
	26	1374.3	2057.9
	27	395.3	31.4
	28	226	18.9
	29	341.8	24.3
	30	329.3	26.7

## Data Availability

Additional data are available from the corresponding author upon request.

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
