# Peer review of "Metallothionein I/II Expression and Metal Ion Levels in Correlation with Amyloid Beta Deposits in the Aged Feline Brain"

_brainsci, 2023, doi:10.3390/brainsci13071115_

Round 1
Reviewer 1 Report
The manuscript by Apostolopoulou EP et al is an extensive study probing different brain regions from cats of different age groups using histopathology histochemistry, immunohistochemistry, and mass spectrometry and describing levels of antioxidant protein metallothionein I/II, metals iron and zinc, and Alzheimer’s amyloid beta deposits in context of aging.
There are some serious issues with the manuscript, which affect the readability of the manuscript and make the conclusion less than convincing.
First of all, the manuscript being a result of a part of thesis, reads more like a thesis rather than a research article, i. e, several sections of the manuscript are too lengthy. For example, the Introduction section, although covers very important and diverse topics related to brain aging, oxidative stress, iron, zinc and metalloprotein, is too long. Running into 9 paragraphs and dwelling across several different concepts, this section seems to be loosing the grip over the reader. It might be worthwhile to re-write this section to make it more concise, to-the-point, and succinct interconnecting various topics covered more seamless. Similarly, the Discussion section is also too lengthy, sometime having redundant discussion.
The most problematic issue with the manuscript is the glaring lack of feline subjects in 2–7 years age group. This leaves a very important age group out of this study. It is not clear why this age group was omitted from this study. Authors should explain the rationale behind not including this cohort in this study. It’s hard to believe that not even a few cats were available to authors in this age group. The whole narrative of the manuscript (i. e., aging-related changes in metallothionein, iron, zinc, amyloid beta and correlation thereof with aging) is incomplete and less convincing in absence of data from this particular age-group missing in their study.
The tabular presentation of data (instead of graphical/chart representation) in the manuscript has caused two problems: First, the data presented in Tables 4, 5, 6, 7, and 8 are % scores without any information on sample size. The % score on its own does not give true representation of variability within the dataThe manuscript by Apostolopoulou EP et al is an extensive study probing different brain regions from cats of different age groups using histopathology histochemistry, immunohistochemistry, and mass spectrometry and describing levels of antioxidant protein metallothionein I/II, metals iron and zinc, and Alzheimer’s amyloid beta deposits in context of aging.
There are some serious issues with the manuscript, which affect the readability of the manuscript and make the conclusion less than convincing.
First of all, the manuscript being a result of a part of thesis, reads more like a thesis rather than a research article, i. e, several sections of the manuscript are too lengthy. For example, the Introduction section, although covers very important and diverse topics related to brain aging, oxidative stress, iron, zinc and metalloprotein, is too long. Running into 9 paragraphs and dwelling across several different concepts, this section seems to be loosing the grip over the reader. It might be worthwhile to re-write this section to make it more concise, to-the-point, and succinct interconnecting various topics covered more seamless. Similarly, the Discussion section is also too lengthy, sometime having redundant discussion.
The most problematic issue with the manuscript is the glaring lack of feline subjects in 2–7 years age group. This leaves a very important age group out of this study. It is not clear why this age group was omitted from this study. Authors should explain the rationale behind not including this cohort in this study. It’s hard to believe that not even a few cats were available to authors in this age group. The whole narrative of the manuscript (i. e., aging-related changes in metallothionein, iron, zinc, amyloid beta and correlation thereof with aging) is incomplete and less convincing in absence of data from this particular age-group missing in their study.
The tabular presentation of data (instead of graphical/chart representation) in the manuscript has caused two problems: First, the data presented in Tables 4, 5, 6, 7, and 8 are % scores without any information on sample size. The % score on its own does not give true representation of variability within the data-set. For example, One out of two observations makes 50% score, while five out of ten observations also make 50% score. However, data is statistically more reliable for larger sample size. In this sense, the percent score for different groups at different grading in Tables 4 through 8 does not reliably represent the histopathology. In addition to above, it is not clear from the tables which groups and which grades were compared for obtaining statistical significance. A better representation of data would be in the form of box-and-whisker plot or stacked bar diagram indicating the data dispersion (and statistical significance) for each brain region.
The manuscript does not convincingly establish correlation between aging and disease features (such as amyloid beta). Most of the pathological signs are present in mature group at grade 1 and/or 2 (and who knows they might have been present even in age-group 2–7 had the data been presented for this cohort). Therefore, it can not be concluded that these features indicate pathological aging. Instead, these indicate simple aging without any disease manifestation. This is particularly true in case of metal ions studied here. Iron and zinc levels were not statistically different among different groups (Section 3.4, Fig 5), the pathologies observed in the study can not be attributed to metal-induced oxidative stress. The whole premise of the study, therefore, become questionable. The data suggest no biometal dyshomeostasis contradicting authors assertion in discussion section (line 491, 559–568).
Some minor points:
Line 118 indicates junior group having 7months to 2yrs old animals, however, Table 1 does not seem to have any animal younger than 1.5 years. Additionally, this group is under-represented in the data-set.
The scoring scores employed in the manuscript for different pathologies were based on number of stained cells/regions. Was it based on certain criteria or just arbitrarily assigned?
The score for Junior group is 0% for all grades for some of the entries in Table 7, 8. What does that mean?
Superscripts on case no. 1, 2, 3 in junior group in Table 1 have not been explained in the footnote.
Line 38, 45, 46 needs an appropriate citation to back the statement.
Line 97, Ref [24] appears to be misquoted.
Minor English problems:
Line 15, "In the present study was assessed"
Line 79, It is not clear what "urgently" indicates here.
Line 479 is a double negative statement.
Minor English check required.
Author Response
We would like to thank the editor and the reviewers for dedicating their time and effort to review the manuscript. We greatly appreciate the thoughtful and valuable comments towards improving our article and accentuating our findings.
The responses to the reviewers’ comments are the following:
Reviewer #1
The manuscript by Apostolopoulou EP et al is an extensive study probing different brain regions from cats of different age groups using histopathology histochemistry, immunohistochemistry, and mass spectrometry and describing levels of antioxidant protein metallothionein I/II, metals iron and zinc, and Alzheimer’s amyloid beta deposits in context of aging.
There are some serious issues with the manuscript, which affect the readability of the manuscript and make the conclusion less than convincing.
First of all, the manuscript being a result of a part of thesis, reads more like a thesis rather than a research article, i. e, several sections of the manuscript are too lengthy. For example, the Introduction section, although covers very important and diverse topics related to brain aging, oxidative stress, iron, zinc and metalloprotein, is too long. Running into 9 paragraphs and dwelling across several different concepts, this section seems to be loosing the grip over the reader. It might be worthwhile to re-write this section to make it more concise, to-the-point, and succinct interconnecting various topics covered more seamless. Similarly, the Discussion section is also too lengthy, sometime having redundant discussion.
Response: We truly comprehend the remark about the length of the manuscript and an effort was made to shorten the Introduction section. Regarding the discussion section, due to the different methods performed and the numerous variables examined, we estimated that a shortened form would make it much more difficult to explain and discuss adequately our results. However, we tried to remove certain sentences without reducing the substantial part.
The most problematic issue with the manuscript is the glaring lack of feline subjects in 2–7 years age group. This leaves a very important age group out of this study. It is not clear why this age group was omitted from this study. Authors should explain the rationale behind not including this cohort in this study. It’s hard to believe that not even a few cats were available to authors in this age group. The whole narrative of the manuscript (i. e., aging-related changes in metallothionein, iron, zinc, amyloid beta and correlation thereof with aging) is incomplete and less convincing in absence of data from this particular age-group missing in their study.
Response: Dear reviewer, thank you for the important annotation. The aim of our study was to investigate the aged feline brain for MT-I/II expression, iron and zinc ion concentration and iron ions distribution as well. Also, an attempt was made to further search for a correlation, positive, none or negative, between these findings and the age-related histopathology, including Aβ deposition. For this reason, our target groups included only aged animals. Feline over 7 years old are considered aged animals and show age-related histopathology such as Αβ deposition, chromatolysis, neuronal necrosis and loss etc (Vite and Head 2014). The 3 animals of Junior group were used as controls. For better understanding the sentence in line 118 (line 106 in the revised manuscript) has been reformed:
“Vogt et al., in 2010 mentioned clustering of cats according to their age in the following four groups: Junior (7 months to 2 years old), Mature (7 to 10 years old), Senior (11–14 years old), and Geriatric (≥15 years old) [32]. In the present study the cats were separated as follows: 12 belonged to the Mature group, 7 to the Senior group and 8 to the Geriatric group. As controls were used the brains of three Junior cats (1.5, 2 and 2 years old).”
The tabular presentation of data (instead of graphical/chart representation) in the manuscript has caused two problems: First, the data presented in Tables 4, 5, 6, 7, and 8 are % scores without any information on sample size. The % score on its own does not give true representation of variability within the data-set. For example, One out of two observations makes 50% score, while five out of ten observations also make 50% score. However, data is statistically more reliable for larger sample size. In this sense, the percent score for different groups at different grading in Tables 4 through 8 does not reliably represent the histopathology. In addition to above, it is not clear from the tables which groups and which grades were compared for obtaining statistical significance. A better representation of data would be in the form of box-and-whisker plot or stacked bar diagram indicating the data dispersion (and statistical significance) for each brain region.
Response: We have changed the data representation and added information on sample size for each variable. Statistical significance was obtained using chi square test analysis in order to assess the differences between the four groups for each variable examined. Furthermore, in all tables, we have added as a footnote the Kruskal Wallis results which shows the statistical differences amongst the individual groups.
The manuscript does not convincingly establish correlation between aging and disease features (such as amyloid beta). Most of the pathological signs are present in mature group at grade 1 and/or 2 (and who knows they might have been present even in age-group 2–7 had the data been presented for this cohort). Therefore, it can not be concluded that these features indicate pathological aging. Instead, these indicate simple aging without any disease manifestation. This is particularly true in case of metal ions studied here. Iron and zinc levels were not statistically different among different groups (Section 3.4, Fig 5), the pathologies observed in the study cannot be attributed to metal-induced oxidative stress. The whole premise of the study, therefore, become questionable. The data suggest no biometal dyshomeostasis contradicting authors assertion in discussion section (line 491, 559–568).
Response: Dear reviewer, as we explained in the previous annotation, we studied only aged animals and used 3 junior cats as controls. Considering that, we described features of normal aging without any brain disease manifestation, since we excluded cases with neurological history and the histopathological findings were associated with age. Aβ deposits have been described in normal aging as well as in neurodegenerative diseases of humans and animals, including cats.
The aim of our study was to investigate the aged feline brain for MT-I/II expression, iron and zinc ion concentration and iron ions distribution as well. Also, an attempt was made to further search for a correlation, positive, none or negative, between these findings and the age-related histopathology, including Aβ deposition. Based on our results we concluded that iron and zinc levels were not statistically significant using ICPMS. However, a mild age-dependent numerical increase of iron and mildly elevated zinc concentration were noticed in the Mature and Senior group compared to those of the Controls and Geriatric groups (fig. 5). Interestingly, statistically significant age-dependent localized increment of iron accumulation was demonstrated using Perl’s/DAB staining regarding the iron plaque load in certain regions as well as myelin-associated iron in the cerebellum. Our findings led us to suggest that there might be a positive correlation between iron and zinc levels as well as MT-I/II expression with Aβ extracellular deposition in the brain of aged cats. Nevertheless, further investigation should be undertaken to confirm our suggestion.
Some minor points:
Line 118 indicates junior group having 7months to 2yrs old animals, however, Table 1 does not seem to have any animal younger than 1.5 years. Additionally, this group is under-represented in the data-set.
Response: Line 118: Dear reviewer, thank you for the important annotation. For better understanding the sentence in line 118 (line 106 in the revised manuscript) has been reformed:
“Vogt et al., in 2010 mentioned clustering of cats according to their age in the following four groups: Junior (7 months to 2 years old), Mature (7 to 10 years old), Senior (11–14 years old), and Geriatric (≥15 years old) [32]. In the present study the cats were separated as follows: 12 belonged to the Mature group, 7 to the Senior group and 8 to the Geriatric group. As controls were used the brains of three Junior cats (1.5, 2 and 2 years old).”
The scoring scores employed in the manuscript for different pathologies were based on number of stained cells/regions. Was it based on certain criteria or just arbitrarily assigned?
Response: The semiquantitative scoring system used for the interpretation of our histopathological and immunohistochemical results were based on previous studies which also examined the same variables and some of them were modified and designed for the present study (Kojima et al. 1999, Brellou et al. 2006, Van Duijn et. al 2017, Boos et al. 2021, Takahashi et al. 2023).
The score for Junior group is 0% for all grades for some of the entries in Table 7, 8. What does that mean?
Response: Please accept our apologies. The tables have been corrected properly.
Superscripts on case no. 1, 2, 3 in junior group in Table 1 have not been explained in the footnote.
Response: The superscripts were accidentally added. Please accept our apologies. They have been removed from the revised manuscript.
Line 38, 45, 46 needs an appropriate citation to back the statement.
Response: We have added appropriate references.
Line 97, Ref [24] appears to be misquoted.
Response: The reference 25 supports that “MT-I/II is upregulated in AD and is primarily associated with Aβ plaques in the hippocampus of several AD animal models” and we used it to report that “Aβ plaques have also been associated with MT-I/II in the hippocampus of experimental animal models of AD”
Minor English problems:
Line 15, "In the present study was assessed»
Response: revised
Line 79, It is not clear what "urgently" indicates here.
Response: revised
Line 479 is a double negative statement.
Response: revised

Reviewer 2 Report
Review of a manuscript “Metallothionein I/II expression and metal ion levels in correlation with amyloid beta deposits in the aged feline brain” by Emmanouela Apostolopoulou and coauthors submitted to the “Brain Sciences”.
Aging of mammalian has been linked with iron and zinc dyshomeostasis and associated with metallothionein I-II expression. These factors are also connected with the pathogenesis of Alzheimer’s disease and other neurodegenerative pathologies. The manuscript is aimed at Investigation of the presence and concentration of iron and zinc, in addition to metallothionein I-II expression in the brain of old animals. This is an important biomedical area and the results resented in the manuscript will be interesting for the readership of the “Brain Sciences”. The following corrections and additions should be made.
Abstract
Line 23. “Aβ immunoreactivity was detected in vessel’s walls, neuronal somata and forming extracellular deposits in cats over 7 years old.” The sentence is clumsy and should be rewritten as follows: ““Aβ immunoreactivity was detected in vessel’s walls and neuronal somata; it is present as extracellular deposits in cats over 7 years old”.
Introduction
Lines 38-39. After the sentence “It has also been described as an important risk factor for many neurodegenerative diseases in humans, including Alzheimer’s disease (AD) and Parkinson’s disease (PD)” the authors should add a reference to the following article [Emamzadeh FN et al., Parkinson’s disease: Biomarkers, Treatment, and Risk Factors. Frontiers in Neuroscience, Neurodegeneration, 12, 61230, August 2018. https://doi.org/10.3389/fnins.2018.00612].
Line 45. “Morphological changes during brain aging contribute to different degrees of behavioral and cognitive impairment.” The authors should add reference after this sentence.
Line 84. “Of these, MT-I and MT-II (hereafter MT-I/II) have been considered equivalent proteins due to their similar structure and similar expression profiles” The term “equivalent proteins” is not completely clear and needs some clarifications. For example, how similar/different are their functions?
Materials and Methods.
Line 113. “The cats died or were deliberately euthanized in the Companion Animal Clinic” Does it mean that some of the animals died from a natural causes?
Results
Figure 1. “Tissue sections from temporal lobes of cats positively stained for GFAP. (a) 414
Grade 0 (Case No 2 aged 2 years old: grade 0 in temporal GM (b) Case No 13 aged 9 years old: Grade 1 in temporal WM (c) Case No 19 aged 11 years old: Grade 2 in temporal WM”
It is unclear why this figure is placed after Figure 2. If this is Figure 3(not 1) the authors should designate it correctly.
Discussion
Line 616. “In conclusion, statistical analysis revealed significant increment of MT-I/II expression in the majority of the variables examined (p <0.1)” The sense of this sentence is unclear. What does it mean in the majority of the variables examined? How this is correlated with aging?. The statement should be explained for easier understanding.
Line 651. “Increased total brain iron has been associated with early Αβ plaque formation in a mouse model of AD [75].” It would be beneficial if the authors explain clearly what changes they found which are specific exclusively for cats and were not described in mice and other animals.
Author Response
We would like to thank the editor and the reviewers for dedicating their time and effort to review the manuscript. We greatly appreciate the thoughtful and valuable comments towards improving our article and accentuating our findings.
The responses to the reviewers’ comments are the following:
Reviewer #2
Abstract
Line 23. “Aβ immunoreactivity was detected in vessel’s walls, neuronal somata and forming extracellular deposits in cats over 7 years old.” The sentence is clumsy and should be rewritten as follows: ““Aβ immunoreactivity was detected in vessel’s walls and neuronal somata; it is present as extracellular deposits in cats over 7 years old”.
Response: Dear reviewer, thank you for the annotation. We would like to notice that all Aβ deposits, in vascular wall, neurons and extracellular were detected in cats over 7 years old. So, we modified the sentence as follows: ““In cats over 7 years old Aβ immunoreactivity was detected in vessel’s walls and neuronal somata; extracellular Aβ deposits were also evident”. We hope this sentence meets your requirement.
Introduction
Lines 38-39. After the sentence “It has also been described as an important risk factor for many neurodegenerative diseases in humans, including Alzheimer’s disease (AD) and Parkinson’s disease (PD)” the authors should add a reference to the following article [Emamzadeh FN et al., Parkinson’s disease: Biomarkers, Treatment, and Risk Factors. Frontiers in Neuroscience, Neurodegeneration, 12, 61230, August 2018. https://doi.org/10.3389/fnins.2018.00612].
Response: We have added the suggested reference.
Line 45. “Morphological changes during brain aging contribute to different degrees of behavioral and cognitive impairment.” The authors should add reference after this sentence.
Response: We have removed this line in our effort to shorten the Introduction section, as proposed by the first reviewer.
Line 84. “Of these, MT-I and MT-II (hereafter MT-I/II) have been considered equivalent proteins due to their similar structure and similar expression profiles” The term “equivalent proteins” is not completely clear and needs some clarifications. For example, how similar/different are their functions?
Response: The sentence has been modified according to your comments.
Materials and Methods.
Line 113. “The cats died or were deliberately euthanized in the Companion Animal Clinic” Does it mean that some of the animals died from a natural causes?
Response: We have modified the sentence as follows “The cats either died by natural causes or traumatic injury or were euthanized in the Companion Animal Clinic (Table 1).”
Results
Figure 1. “Tissue sections from temporal lobes of cats positively stained for GFAP. (a) 414
Grade 0 (Case No 2 aged 2 years old: grade 0 in temporal GM (b) Case No 13 aged 9 years old: Grade 1 in temporal WM (c) Case No 19 aged 11 years old: Grade 2 in temporal WM”
It is unclear why this figure is placed after Figure 2. If this is Figure 3(not 1) the authors should designate it correctly.
Response: Revised
Discussion
Line 616. “In conclusion, statistical analysis revealed significant increment of MT-I/II expression in the majority of the variables examined (p <0.1)” The sense of this sentence is unclear. What does it mean in the majority of the variables examined? How this is correlated with aging? The statement should be explained for easier understanding.
Response: The variables examined regarding MT-I/II expression are displayed in Table 6. All the variables except MT-I/II immunoreactivity in temporal and frontal leptomeningeal, GM and WM blood vessels were statistically significant (p< 0.1). Taking into consideration that we compared different age groups using the chi square test, it might be concluded that there is an age-related expression of MT-I/II in our study. For better understanding, we modified the sentence and explained it in detail.
Line 651. “Increased total brain iron has been associated with early Αβ plaque formation in a mouse model of AD [75].” It would be beneficial if the authors explain clearly what changes they found which are specific exclusively for cats and were not described in mice and other animals.
Response: Our results are in accordance with previous studies in mice and other animal species which have been mentioned in our manuscript (Morita et al. 2005, Kim et al. 2005, Meguro et al. 2008, Sands et al. 2016, Mezzanote et al. 2022). We revised the lines 572-573 and 582-583 (in the revised manuscript). We have highlighted that “iron myelin-associated staining was also observed for the first time in the aging feline brain located in the frontal, temporal and cerebellar white matter, thalamus, and hippocampal fimbria” and we demonstrated that “the nucleolus was the primary iron labelled site in the aged feline brain“.

Round 2
Reviewer 1 Report
In their revised manuscript, the authors have adequately addressed all the concern raised by this reviewer.
Minor English editing may be needed.